# Stimulation of the catalytic activity of the tyrosine kinase Btk by the adaptor protein Grb2

**Laura M Nocka[1,2,3]†, Timothy J Eisen[1,2,4], Anthony T Iavarone[2,5], Jay T Groves[1,2,6]\*, John Kuriyan[1,2,3,4]\*‡**

[1]Department of Chemistry, University of California, Berkeley, Berkeley, United States; [2]California Institute for Quantitative Biosciences, University of California, Berkeley, Berkeley, United States; [3]Howard Hughes Medical Institute, University of California, Berkeley, Berkeley, United States; [4]Department of Molecular and Cell Biology, University of California, Berkeley, Berkeley, United States; [5]College of Chemistry Mass Spectrometry Facility, University of California, Berkeley, Berkeley, United States; [6]Institute for Digital Molecular Analytics and Science, Nanyang Technological University, Singapore, Singapore

**\*For correspondence:**
JTGroves@lbl.gov (JTG);
john.kuriyan@vanderbilt.edu (JK)

**Present address:** †Department of Biochemistry, Stanford University School of Medicine, Stanford, United States; ‡Department of Biochemistry, Vanderbilt University School of Medicine, Nashville, United States

**Abstract** The Tec-family kinase Btk contains a lipid-binding Pleckstrin homology and Tec homology (PH-TH) module connected by a proline-rich linker to a 'Src module', an SH3-SH2-kinase unit also found in Src-family kinases and Abl. We showed previously that Btk is activated by PH-TH dimerization, which is triggered on membranes by the phosphatidyl inositol phosphate $PIP_3$, or in solution by inositol hexakisphosphate ($IP_6$) (Wang et al., 2015, https://doi.org/10.7554/eLife. 06074). We now report that the ubiquitous adaptor protein growth-factor-receptor-bound protein 2 (Grb2) binds to and substantially increases the activity of $PIP_3$-bound Btk on membranes. Using reconstitution on supported-lipid bilayers, we find that Grb2 can be recruited to membrane-bound Btk through interaction with the proline-rich linker in Btk. This interaction requires intact Grb2, containing both SH3 domains and the SH2 domain, but does not require that the SH2 domain be able to bind phosphorylated tyrosine residues – thus Grb2 bound to Btk is free to interact with scaffold proteins via the SH2 domain. We show that the Grb2-Btk interaction recruits Btk to scaffold-mediated signaling clusters in reconstituted membranes. Our findings indicate that $PIP_3$-mediated dimerization of Btk does not fully activate Btk, and that Btk adopts an autoinhibited state at the membrane that is released by Grb2.

## Editor's evaluation

This important study reports an unexpected mode of activation of the critical immune cell kinase, Btk, by the SH3-SH2 domain-containing adaptor protein Grb2. The authors convincingly demonstrate that Grb2 binding to a Pro-rich region positioned between the Btk PH-TH domains and Src module plays a crucial role in relieving Btk autoinhibition and mediating Btk recruitment into signaling clusters on artificial membranes. These findings provide a mechanistic basis for the clusters previously reported in cells, and resolve in part why Btk can be activated via modes distinct from its close relatives, such as Itk kinase.

## Introduction

B-cell signaling relies on the sequential activation of three tyrosine kinases that transduce the stimulation of the B-cell receptor (BCR) into the generation of calcium flux and the initiation of signaling pathways (*Engels et al., 2008*; *Weiss and Littman, 1994*). The first two kinases in the sequence of activating events are a Src-family kinase, Lyn, and a Syk/ZAP70-family kinase, Syk. The third kinase in this pathway is the Tec-family kinase Btk, which is critical for the development and proliferation of mature B cells (*Figure 1A*; *Hendriks et al., 2014*; *Rip et al., 2018*; *Scharenberg et al., 2007*; *Tsukada et al., 1993*).

Btk has an N-terminal PH-TH module, consisting of a Pleckstrin-homology (PH) domain followed by a Zinc-bound Tec-homology (TH) domain. The PH-TH module is connected through a proline-rich linker to a Src module, comprising an SH3 domain, an SH2 domain, and a kinase domain (*Figure 1B*; *Joseph et al., 2017*; *Shah et al., 2018*; *Wang et al., 2015*). The Src module of Btk is structurally similar to the corresponding modules of other cytoplasmic tyrosine kinases, such as c-Src, Lck, and Abl. In the autoinhibited form, the SH3 domain and the N-terminal lobe of the kinase domain sandwich the SH2-kinase linker, and the SH2 domain sits adjacent to the C-terminal lobe of the kinase domain (*Figure 1B*). Together these contacts prevent the kinase domain from adopting an active conformation (*Amatya et al., 2019*; *Shah et al., 2018*). The PH-TH module of Btk is connected to the Src module by a 44-residue linker that contains ten proline residues and two PxxP motifs that are potential SH3-binding sites (*Figure 1B*; *Joseph et al., 2017*; *Shah et al., 2018*; *Wang et al., 2015*). Little is known about the role of this linker, though other Tec-family kinases also contain at least one PxxP motif in this linker.

Crystal structures of the Btk PH-TH module, first determined by the late Matti Saraste and colleagues, reveal a dimeric arrangement, which we refer to as the "Saraste dimer" (*Baraldi et al., 1999*; *Hyvönen and Saraste, 1997*; *Figure 1C*). Mutagenesis of residues at the Saraste-dimer interface has shown that the PH-TH dimer is critical for Btk activity in cells (*Chung et al., 2019*). Btk is activated in vitro by vesicles containing $PIP_3$, or in solution by inositol hexakisphosphate ($IP_6$) (*Kim et al., 2019*; *Wang et al., 2015*). Experiments on supported-lipid bilayers, as well as molecular dynamics simulations, have shown that $PIP_3$ in membranes promotes dimerization of the PH-TH module through binding to at least two sites (*Chung et al., 2019*; *Wang et al., 2019*).

Growth-factor-receptor-bound protein 2 (Grb2) is a ubiquitous adaptor protein that is important for receptor tyrosine kinase signaling and the activation of the mitogen-activated protein kinase (MAPK) pathway (*Cantor et al., 2018*; *Clark et al., 1992*; *Gale et al., 1993*; *Lowenstein et al., 1992*; *Olivier et al., 1993*). Grb2 consists of two SH3 domains that flank an SH2 domain (*Figure 1D*). Grb2 is responsible for bringing signaling enzymes to their substrates and also helps in the formation of signaling clusters at the plasma membrane by virtue of its ability to crosslink scaffold proteins or receptors (*Huang et al., 2019*; *Huang et al., 2017a*; *Huang et al., 2016*; *Lin et al., 2022*; *Su et al., 2016*). For example, in the MAPK pathway, the SH3 domains of two Grb2 molecules bind to the Ras activator Son of Sevenless (SOS). Grb2 binds to scaffold proteins, such as SLP65 (also known as B cell linker protein, BLNK) in B cells and LAT (Linker for Activation of T cells) in T cells, through interaction of the Grb2 SH2 domain with phosphotyrosine residues on the scaffold proteins (*Engels et al., 2009*; *Reif et al., 1994*). In this way, Grb2 recruits SOS to the scaffold proteins or receptors, and the interaction of one SOS molecule with two Grb2 molecules creates a bridge between scaffold proteins. In T cells, a protein closely related to Grb2, GADS (Grb2-related adaptor downstream of Shc), works together with Grb2 to crosslink scaffolding proteins and enzymes downstream of the TCR (T-cell receptor) (*Liu et al., 1999*). Grb2 is also reported to be capable of dimerization, and dimeric Grb2 hinders basal signaling of the fibroblast growth factor receptor 2 (FGFR2), thereby tuning the activity of the receptor (*Lin et al., 2012*; *Sandouk et al., 2023*).

Memory-type B cells expressing membrane-bound IgG (mIgG) as a component of the BCR rely on an immunoglobulin tail tyrosine (ITT) motif that is phosphorylated by Syk and recruits Grb2 through binding of its SH2 domain to the resulting phosphotyrosine residues. It was shown, by fusing domains of Grb2 to the IgG tail (in the absence of the ITT motif), that the N-terminal SH3 domain of Grb2 plays a critical role in generating downstream signals. Immunoprecipitation using the isolated N-terminal SH3 domain of Grb2 showed that this domain interacts with Btk, as well as with the ubiquitin ligase Cbl and the GTPase activator SOS (*Engels et al., 2014*). Substitution of the N-terminal SH3 domain of Grb2 by the C-terminal SH3 domain of Grb2, or by the N-terminal SH3 domain of the Grb2 family

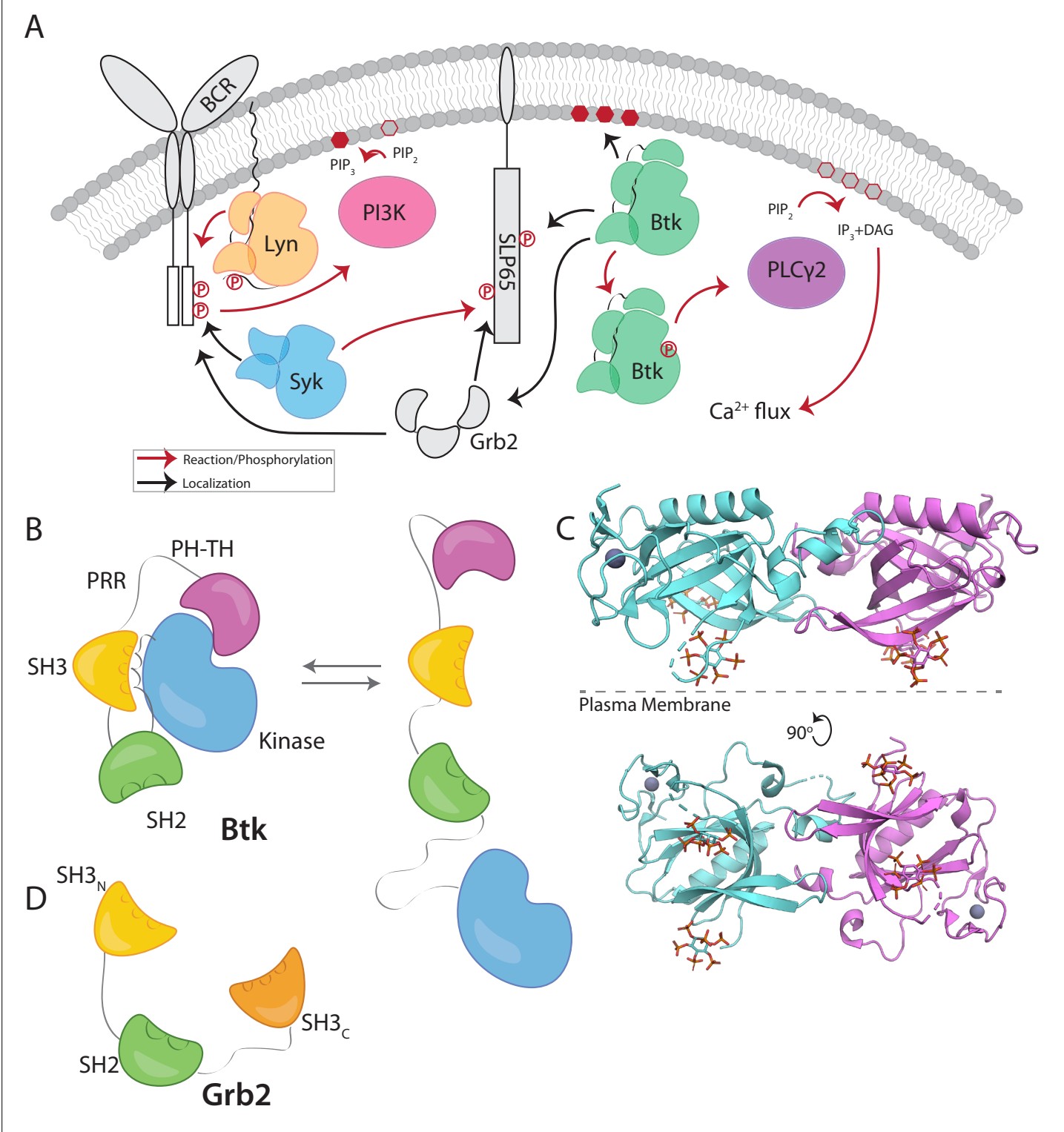

**Figure 1.** Btk and Grb2 are integral to B-cell signaling. (**A**) The B-cell signaling cascade relies on kinases and adaptor molecules. (**B**) Btk is made up of five distinct domains. The Pleckstrin homology and Tec homology domains fold together into one module (PH-TH), which is connected through a linker containing two proline-rich regions (PRR) to the Src module (SH3, SH2, and Kinase domain). These domains together mediate autoinhibition of Btk. (**C**) Crystal structure of the Saraste dimer of the Btk PH-TH module, with each module bound to two inositol hexakisphosphate (IP$_6$) molecules, mimicking PIP$_3$ binding (PDB: 4Y94). (**D**) Grb2 domain architecture consists of two SH3 domains that flank an SH2 domain.

member GRAP resulted in failure to trigger $Ca^{2+}$ flux. In the mIgG fusion with the N-terminal SH3 domain of GRAP, $Ca^{2+}$ flux was restored by the substitution of three residues in the GRAP SH3 domain by the corresponding residues in Grb2 (*Engels et al., 2014*). Thus, an interaction between Grb2 and Btk plays a critical role in BCR signaling.

We now report the discovery, made using in-vitro reconstitution of Btk on $PIP_3$-containing membranes, that Btk and Grb2 interact through the proline-rich linker of Btk, and that, through this interaction, Grb2 stimulates the kinase activity of Btk. The activation of Btk by Grb2 relies primarily on the N-terminal SH3 domain of Grb2 and does not require that the SH2 domain be able to bind to phosphotyrosine residues – we infer that the SH2 domain is free to interact with phosphotyrosine residues presented by scaffold proteins. Our data suggest a mechanism whereby the binding of one SH3 domain of Grb2 to the proline-rich linker of Btk disrupts Btk autoinhibition. Thus, the interaction between Grb2 and Btk can integrate the localization of Btk with its activation.

## Results and discussion

### Btk recruits Grb2 to $PIP_3$-containing supported-lipid bilayers through the proline-rich region of Btk

To probe the interaction between Btk and Grb2, we utilized a supported-lipid bilayer system that we had used previously to characterize the interaction of the isolated Btk PH-TH module with lipids (*Chung et al., 2019*). Using this system we discovered, unexpectedly, that Grb2 can be recruited to the membrane via an interaction with membrane-bound full-length Btk. We had shown previously that the PH-TH module of Btk is recruited to membranes containing 4% $PIP_3$, and that this recruitment exhibits a sharp dependence on $PIP_3$ concentration in the membrane. To characterize the interaction between Grb2 and full-length Btk, we used supported-lipid bilayers containing 4% $PIP_3$, to which full-length Btk is recruited from solution via the PH-TH module (*Chung et al., 2019*). We found that recruitment of Btk to the membrane also resulted in Grb2 being recruited to the membrane (*Figure 2A* and *Figure 2—figure supplement 1*).

Recruitment of Grb2 to the membrane was measured using total internal reflection fluorescence (TIRF) microscopy and Grb2 labeled with Alexa Fluor 647 (Grb2-647) through maleimide coupling. Grb2 contains multiple surface cysteines, and therefore labeling was carried out at 1–2-fold molar excess to ensure that at most one cysteine was labeled on any molecule of Grb2. This resulted in about 17% labeling overall. TIRF imaging provides a highly selective measurement of membrane-associated Grb2-647, without picking up signal from protein in the solution phase (*Huang et al., 2017a*). When Btk and Grb2-647 were added together, Grb2-647 was recruited to the bilayer, as indicated by an increase in fluorescence (*Figure 2A* and *Figure 2—figure supplement 1*). When Grb2-647 was added to the supported-lipid bilayers without Btk there was no change in fluorescence above background (*Figure 2A*). These experiments indicate that Grb2 is capable of binding directly to Btk in the absence of other proteins.

We next addressed the question of which regions of Btk are necessary for the interaction with Grb2. To do this, we tethered various constructs of Btk to membranes containing DGS-NTA(Ni) (1,2-dioleoyl-sn-glycero-3-[(N-(5-amino-1-carboxypentyl)iminodiacetic acid)succinyl] (nickel salt)) lipids by using an N-terminal hexa-histidine tag on Btk, rather than relying on the binding of the PH-TH module to $PIP_3$. Constructs of N-terminally His-tagged Btk could then be tethered directly to the membrane through the binding of the histidine tag to DGS-NTA(Ni) lipids. This tethering method has been shown to result in limited unbinding of the His-tagged protein over the timescale of our experiments (less than one hour) (*Nye and Groves, 2008*). This allowed us to study the binding of Grb2 to constructs of Btk that do not contain the PH-TH module (see Materials and Methods for precise definition of the Btk and Grb2 constructs).

Supported-lipid bilayers containing 4% DGS-NTA(Ni) were prepared and His-tagged Btk constructs were added to these bilayers at different concentrations. Each of the following constructs was tested by adding Grb2-647 and monitoring the change in TIRF intensity at the membrane: full-length Btk, Btk in which the PH-TH module and proline-rich linker are deleted (SH3-SH2-kinase; residues 212–659 of human Btk), SH2-kinase (residues 281–659), the kinase domain alone (residues 402–659), and the proline-rich linker alone (residues 171–214). Grb2-647 is recruited to the bilayer when full-length Btk or the isolated proline-rich linker are tethered to the membrane. The tethering of other constructs to

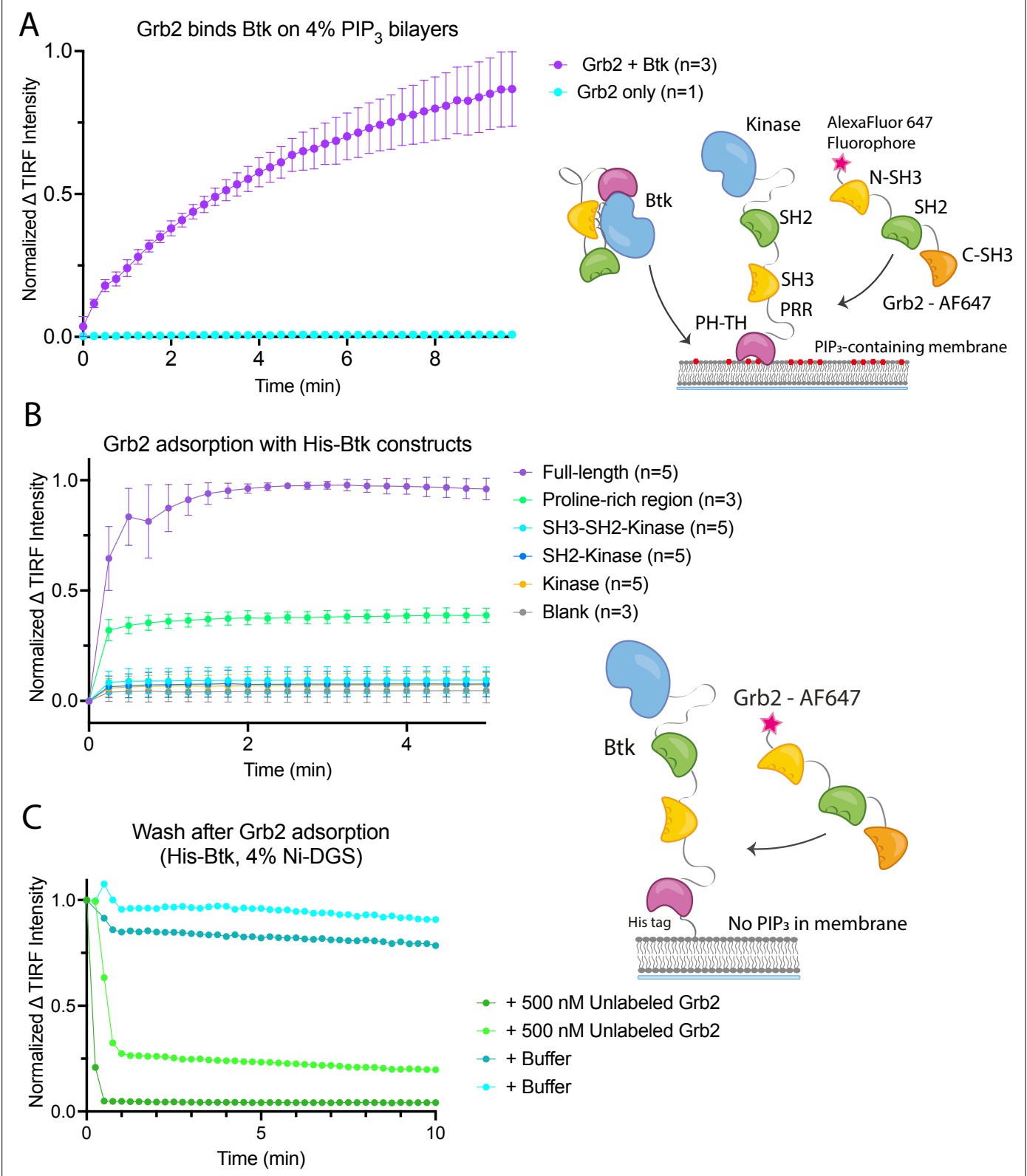

**Figure 2.** Grb2 interacts with membrane-bound Btk. (**A**) Fluorescently labeled Grb2 was added to supported-lipid bilayers containing 4% PIP$_3$, with (n=3) or without (n=1) Btk. The change in fluorescence intensity, monitored by TIRF, is plotted over time. (**B**) Various constructs of Btk [Btk (n=5), SH3-SH2-Kinase (n=5), SH2-Kinase (n=5), Proline-rich linker (n=3), Kinase (n=5), Blank (no protein, n=3)] with His tags were tethered to supported-lipid bilayers containing 4% DGS-NTA(Ni) lipids. The bilayers were washed after equilibration to remove any weakly bound Btk. Fluorescently-labeled Grb2

*Figure 2 continued on next page*

*Figure 2 continued*

was added to these bilayers and the change in fluorescence intensity with time is shown. (**C**) The binding of Grb2 to membrane bound Btk is reversible. Grb2 was added to a bilayer decorated with His-tagged Btk. In one experiment (blue lines, n=2), the bilayer was washed with buffer, and in another experiment (green lines, n=2), the bilayer was washed with a solution containing 10-fold higher concentration of unlabeled Grb2. All error bars in this figure represent standard deviation across replicates. See *Figure 2—source data 1* for TIRF intensity data for A-C, and Dryad repository for raw image files, Figure 2—source data 2 (https://dx.doi.org/10.5061/dryad.8sf7m0ctd).

The online version of this article includes the following source data and figure supplement(s) for figure 2:

**Source data 1.** Raw image files used to generate *Figure 2A–C*.

**Figure supplement 1.** Grb2 adsorption scales with Btk concentration.

the membrane did not show an intensity change above background upon addition of labeled Grb2 (*Figure 2B*). We confirmed that the observed fluorescence change was not due to an irreversible process, such as protein aggregation, by showing that the increase in fluorescence could be reversed rapidly by addition of unlabeled Grb2 at a fivefold higher solution concentration than labeled Grb2 (*Figure 2C*).

An alternative, plausible mechanism for Grb2 recruitment to Btk is interaction between the SH2 domain of Grb2 and a phosphotyrosine residue on Btk. However, the Btk constructs used in these experiments were expressed using a bacterial system in which co-expression of a tyrosine phosphatase is expected to maintain the proteins in the unphosphorylated state, despite the presence of the Btk kinase domain in some of the constructs (*Seeliger et al., 2005*; *Wang et al., 2015*). We confirmed that full-length Btk was not phosphorylated using western blot analysis with a pan-phosphotyrosine antibody. Additionally, we did not observe any phosphorylated peptides by mass spectrometry, ruling out that the interaction depends on an SH2-phosphotyrosine interaction.

These experiments demonstrate that the proline-rich linker of Btk is able to recruit Grb2 to the membrane, without the other domains of Btk being present. No binding is detected to constructs that lack the PH-TH module and the proline-rich linker. We conclude that the proline-rich linker is a principal determinant of the interaction between Btk and Grb2.

## Grb2 enhances the kinase activity of Btk

Btk is activated by PIP$_3$-containing vesicles, as shown by experiments in which the phosphorylation of full-length Btk was monitored by western blot with pan-phosphotyrosine antibody (*Wang et al., 2015*). We repeated those experiments by incubating Btk (1 µM bulk solution concentration) in the presence or absence of lipid vesicles containing 4% PIP$_3$, and then added increasing concentrations of Grb2, from 0 to 10 µM bulk solution concentration. In the absence of vesicles, no change in phosphorylation is detected when Grb2 is added to Btk (*Figure 3A–B*, and *Figure 3—figure supplement 1*). In the presence of PIP$_3$-containing vesicles, the addition of Grb2 results in substantially increased levels of Btk phosphorylation, compared to the presence of PIP$_3$-containing vesicles alone. When Btk is mixed with Grb2, the phosphorylation level detected at 5 min or 20 min for Btk without Grb2 (*Figure 3A–B* and *Figure 3—figure supplement 1*).

We also tested whether the binding of Grb2 to Btk influences the ability of Btk to phosphorylate its specific substrate, PLCγ2. To do this, we monitored phosphorylation of a peptide segment spanning residues 746–766 of PLCγ2 that contains two tyrosine residues (Tyr 753 and Tyr 759). Phosphorylation of this segment by Btk plays a key role in the activation of PLCγ2 in B cells (*Ozdener et al., 2002*; *Rodriguez et al., 2001*). This peptide segment was fused to an N-terminal SUMO protein and green fluorescent protein (GFP) (referred to as the PLCγ2-peptide fusion) to allow for visualization on a gel, as the substrate is otherwise too small to analyze by western blot. A protein in which the SUMO tag was cleaved is also included. We titrated Grb2 from 0 to 10 µM along with 1 µM Btk and 10 µM PLCγ2-peptide fusion, with or without 250 µM 4% PIP$_3$-containing vesicles (*Figure 3C* and *Figure 3—figure supplement 2*). Additionally, we added 1 µM Btk to 250 µM 4% PIP$_3$-containing vesicles in the presence of 10 µM PLCγ2-peptide fusion, with or without the addition of Grb2 (*Figure 3D* and *Figure 3—figure supplement 3*). Phosphorylation of PLCγ2-peptide fusion was measured by western blot analysis of total phosphotyrosine, using a pan-phosphotyrosine antibody. Under these conditions, the presence of Grb2 enhances phosphorylation of the PLCγ2-peptide fusion substantially (*Figure 3C–D* and *Figure 3—figure supplements 2 and 3*). We used mass spectrometry to confirm that both tyrosine residues, Tyr 753 and Tyr 759, within the PLCγ2 peptide were phosphorylated

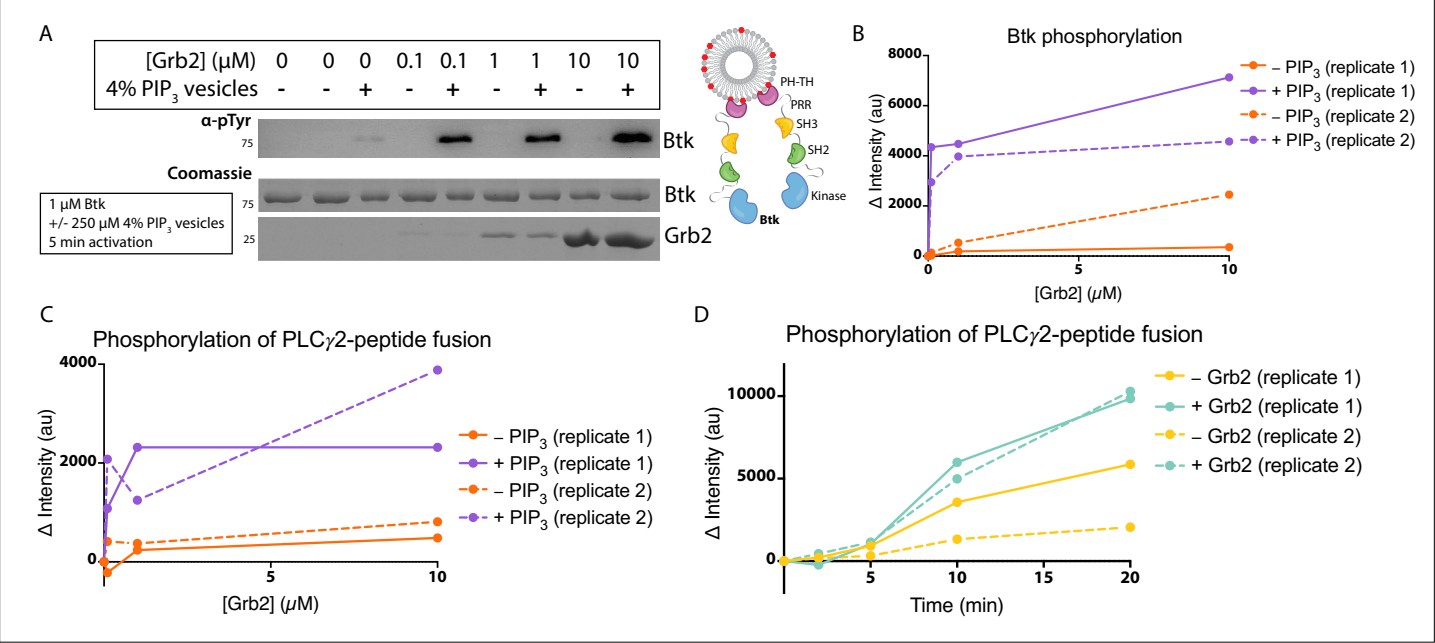

**Figure 3.** Grb2 enhances Btk kinase activity in a $PIP_3$-dependent manner. The activity of Btk is monitored by measuring the phosphorylation of Btk, Grb2, and a PLCγ2-peptide fusion protein by western blot. (**A**) Grb2 was titrated into samples containing Btk with or without 4% $PIP_3$ lipid vesicles. These samples were activated for 5 min and then quenched with 100 mM EDTA. Total phosphorylation was measured by western blot with pan-phosphotyrosine antibody. See *Figure 3—figure supplement 1* for an independent experimental replicate, and *Figure 3—source data 1*. (**B**) Btk phosphorylation measured from (**A**) along with one independent experimental replicate (*Figure 3—figure supplement 1*). (**C**) Phosphorylation of PLCγ2-peptide fusion by Btk. See blots in *Figure 3—figure supplement 2* for raw data. Grb2 is titrated with or without $PIP_3$, and phosphorylation of PLCγ2-peptide fusion is measured. Change in intensity is determined by comparison to the intensity observed for 0 nM Grb2 for either case (+/−Grb2). See *Figure 3—source data 1* for raw images. (**D**) Measure of PLCγ2-peptide fusion phosphorylation over time with or without the addition of Grb2. All samples contain Btk, PLCγ2-peptide fusion and 4% $PIP_3$ lipid vesicles. Samples were activated and quenched at the timepoints listed (0–20 min). See *Figure 3—figure supplement 3* for raw data and *Figure 3—source data 1*.

The online version of this article includes the following source data and figure supplement(s) for figure 3:

**Source data 1.** Raw images of gels and western membranes referenced in *Figure 3*.

**Figure supplement 1.** Grb2 enhances *trans*-autophosphorylation of Btk in the presence of $PIP_3$ containing vesicles.

**Figure supplement 2.** Grb2 enhances phosphorylation of PLCγ2 by Btk in the presence of $PIP_3$ containing vesicles.

**Figure supplement 3.** Grb2 enhances phosphorylation of PLCγ2 by Btk over time.

**Figure supplement 4.** Phosphorylation of PLCγ2-peptide fusion Tyr corresponding to PLCγ2 Tyr 753 by Btk in the presence of Grb2.

**Figure supplement 5.** Phosphorylation of PLCγ2-peptide fusion Tyr corresponding to PLCγ2 Tyr 759 by Btk in the presence of Grb2.

---

(*Figure 3—figure supplements 4 and 5*). Additionally, a tyrosine residue at the C-terminus of GFP was detected as phosphorylated.

## All three domains of Grb2 are necessary for stimulation of Btk kinase activity

The observed stimulation of Btk catalytic activity by Grb2 prompted us to ask which domains of Grb2 are required for this phenomenon. We made constructs corresponding to each individual domain of Grb2 (N-terminal SH3, SH2, and C-terminal SH3), or combinations of domains (N-terminal SH3-SH2, SH2-C-terminal SH3, and N-terminal SH3-C-terminal SH3, see Materials and Methods for the specification of these constructs). We included an additional Grb2 construct, R86K, in which a conserved arginine residue in the SH2 domain that is critical for phosphotyrosine binding is mutated to lysine. Substitution of the corresponding arginine residue by lysine in other SH2 domains attenuates the binding of the SH2 domains to phosphorylated peptides (*Mayer et al., 1992*). We have demonstrated recently that the R86K mutation impairs the ability of Grb2 to promote phase separation of scaffold proteins (*Lin et al., 2022*). Another Grb2 variant (Y160E) has a reduced capacity for dimerization (*Ahmed et al., 2015*). The ability of each of these constructs to stimulate Btk activity was tested, as

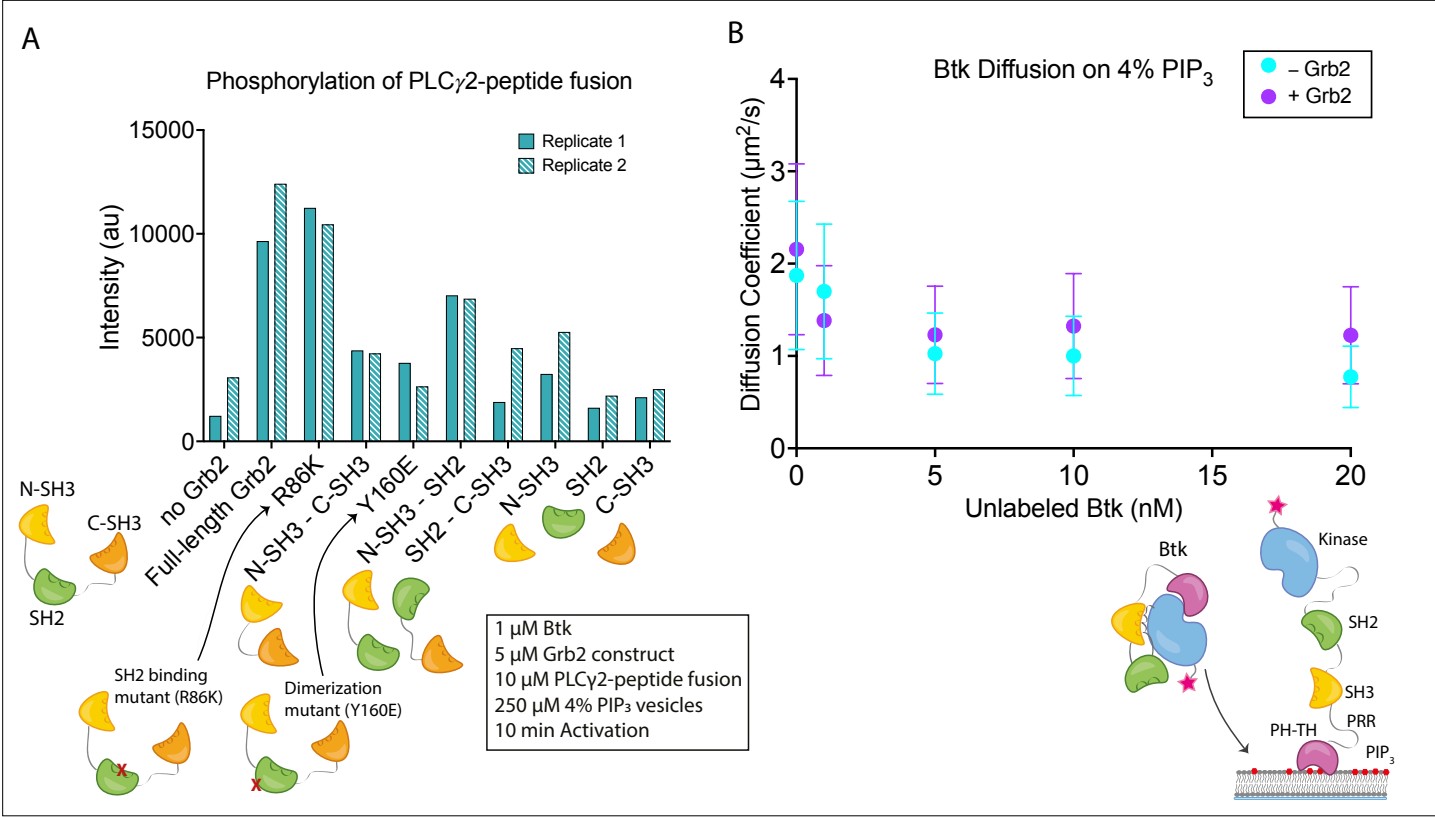

**Figure 4.** Effects of Grb2 on catalytic activity and membrane dynamics of Btk. (**A**) Quantification of PLCγ2-peptide fusion phosphorylation measured from two independent experimental replicates. Phosphorylation of PLCγ2-peptide fusion by Btk in the presence of 4% PIP₃ lipid vesicles was monitored in the presence of various Grb2 constructs. These constructs include: full length Grb2, R86K, N-terminal SH3 fused to C-terminal SH3, Y160E, N-terminal SH3-SH2, SH2-C-terminal SH3, N-terminal SH3, SH2, and C-terminal SH3. See *Figure 4—figure supplement 1* and *Figure 4—source data 1* for raw images. (**B**) Diffusion coefficients for membrane bound Btk were determined by single-molecule tracking, and fitting to step-size distributions for each concentration of unlabeled Btk, with or without Grb2. Error bars represent the standard deviation determined from the fit step-size distribution. See Dryad repository for source data for this panel, Figure 4B – source data (https://dx.doi.org/10.5061/dryad.jwstqjqfd).

The online version of this article includes the following source data and figure supplement(s) for figure 4:

**Source data 1.** Raw images of gels and western membranes referenced in *Figure 4A*.

**Figure supplement 1.** Phosphorylation of PLCγ2-peptide fusion by Btk with the addition of various Grb2 constructs.

**Figure supplement 2.** The diffusion of Btk on membranes in the absence and presence of Grb2.

**Figure supplement 3.** Btk dwell time confirms that Grb2 does not impact Btk membrane dynamics.

measured by phosphorylation of the PLCγ2-peptide fusion (*Figure 4A* and *Figure 4—figure supplement 1*). All reactions were carried out in the presence of unilamellar vesicles at a total lipid concentration of 250 µM containing 4% PIP₃. Protein and vesicles were incubated together for 15 min in the absence of ATP, and the reaction time is measured from when ATP was added to the solution.

Full-length Grb2 increases the phosphorylation of the PLCγ2-peptide fusion by five to six-fold relative to the reaction in which no Grb2 was added (*Figure 4A* and *Figure 4—figure supplement 1*). The Grb2 variant in which the ability of the SH2 domain to bind to phosphopeptides is impaired (Grb2 R86K) stimulates the reaction to essentially the same extent as wild-type Grb2, indicating that the phosphopeptide-binding ability of the Grb2 SH2 domain is not required for stimulation of Btk activity by Grb2. This observation suggests that Grb2 bound to Btk will retain the capacity to dock on phosphorylated scaffold proteins via the SH2 domain, as discussed below. Removal of any of the three component domains of Grb2 results in substantial reduction of phosphorylation of the PLCγ2-peptide fusion (*Figure 4A* and *Figure 4—figure supplement 1*). Grb2-Y160E shows a reduction in the PLCγ2-peptide fusion phosphorylation by Btk, compared to the wild-type Grb2. This result indicates

that Grb2 dimerization may be important for stimulation of Btk activity, although a definitive analysis of the mechanism awaits further study of additional Grb2 mutants.

Constructs containing only the N-terminal SH3 domain of Grb2 increase phosphorylation to a greater extent than constructs containing only the C-terminal SH3 domain (*Figure 4A* and *Figure 4— figure supplement 1*). This indicates that the N-terminal SH3 domain of Grb2 is more important for the interaction with Btk, consistent with the earlier finding that fusion of the N-terminal SH3 domain of Grb2 to mIgG is sufficient for promoting Ca$^{2+}$ flux through the activation of Btk (*Engels et al., 2014*).

## Grb2 does not affect dimerization of Btk on the membrane

Btk activity is stimulated by homodimerization of the PH-TH modules (*Chung et al., 2019*; *Wang et al., 2015*). Given this, it is possible that Grb2 could promote activation by crosslinking two Btk molecules. This would be similar in mechanism to the recent findings that show activation of Btk through dimerization of HIV Nef (*Aryal et al., 2022*). To check whether Grb2 impacts the dimerization of Btk we measured the diffusion coefficient and the dwell time of Btk on the membrane in the presence or absence of Grb2 (*Figure 4B* and *Figure 4—figure supplements 2 and 3*; *Chung et al., 2019*). Although there is no simple relation connecting two-dimensional diffusion and molecular complex size as there is for three-dimensional diffusion, two-dimensional diffusion on a membrane surface nonetheless changes markedly between monomers and dimers and is a sensitive measurement of dimerization (*Chung et al., 2018*; *Chung et al., 2019*; *Kaizuka and Groves, 2004*; *Knight and Falke, 2009*). If Grb2 increases the population of Btk dimers, we would expect to see a decrease in the diffusion coefficient of individual complexes on the membrane in the presence of Grb2. If Grb2 increases the affinity of Btk for the membrane, this would be manifested as an increase in Btk dwell time, the time that single molecules of Btk stay at the membrane, through a reduction in off-rate.

To enable site-specific labeling of full-length Btk we used unnatural amino acid incorporation of an azido-phenylalanine (AzF) group on the surface of the kinase domain of Btk (*Chatterjee et al., 2013*; *Chin et al., 2002*). AzF enables the use of an azide-reactive dye to label the protein and eliminates non-specific labeling at other sites (see Materials and Methods for details). Several sites were tested, and incorporation of AzF at position 403 (Thr 403 in wild-type Btk) showed the best yield of labeled Btk. Thr 403 is a surface-exposed sidechain in the N-lobe of the kinase domain, and we do not anticipate that incorporation of AzF at this position will disturb the structure of Btk. For all data utilizing fluorescent full-length Btk, the construct used is Btk T403AzF labeled with azide reactive Cy5 (Btk-Cy5).

The surface density of Btk-Cy5 on the membrane was observed to increase when the solution concentration of unlabeled Btk was increased from 0 nM to 20 nM, in the presence of very low concentrations of Btk-Cy5 (500 pM-1 nM). This enables the monitoring of single molecules of Btk-Cy5, which was done either in the presence of Grb2 at 50 nM bulk concentration, or without Grb2. From each sample, step-size distributions were compiled from the single-molecule trajectories to assess the various time-dependent components to Btk diffusion on the membrane. Step-size distributions generally required fitting to three components, while two-component exponential fits were sufficient for fitting dwell time. Both with and without Grb2, the fastest of the three diffusion constants decreases with increasing Btk concentration, indicative of a transition from monomeric to dimeric Btk. The presence of Grb2 does not change the diffusive behavior of Btk at any of the concentrations used in these experiments (*Figure 4B* and *Figure 4—figure supplement 2*). This is supported by the same trend observed through dwell time measurement: overall dwell time increases with increasing Btk concentration, but Grb2 has no influence on this change (*Figure 4—figure supplement 3*). The decrease in the Btk diffusion constant with increasing solution concentration of Btk is consistent with what is observed for the Btk PH-TH module at these concentrations, confirming that full-length Btk interacts with the membrane in the same way as does the Btk PH-TH module (*Figure 4B* and *Figure 4— figure supplement 2*; *Chung et al., 2019*). These experiments show that Grb2 does not change the dynamics of membrane-bound Btk, either through changes in the dimer population or changes in the membrane affinity.

## Grb2 can recruit Btk to clusters of scaffold proteins

We studied how the ability of Grb2 to bind to Btk might impact the localization of Btk on the membrane, by monitoring the interaction of Btk with the scaffold protein LAT on supported-lipid bilayers. LAT is

similar to the B-cell scaffolding protein SLP65/BLNK. Our use of LAT, rather than SLP65/BLNK, was predicated on our extensive prior work with LAT on supported-lipid bilayers (*Hashimoto et al., 1999*; *Huang et al., 2019*; *Huang et al., 2017a*; *Huang et al., 2017b*; *Huang et al., 2016*; *Koretzky et al., 2006*; *Su et al., 2016*).

LAT signaling clusters can be generated from minimal components on supported-lipid bilayers. Bilayers containing 4% DGS-NTA(Ni) were prepared with both LAT and the Src-family kinase Hck tethered to the membrane through His-tags. LAT was phosphorylated by Hck, as described previously, before other components were added (*Huang et al., 2019*; *Huang et al., 2016*; *Huang et al., 2017b*). The diffusion constants for phosphorylated LAT and Hck are similar to those for the lipids under these conditions, indicating lack of clustering on their own. Upon the addition of Grb2 and the proline-rich region of the Ras activator SOS (SOS-PRR), the phosphorylated LAT undergoes a protein-condensation phase transition and forms gel-like domains of protein-rich areas in which LAT no longer diffuses freely (*Huang et al., 2017b*; *Su et al., 2016*). This phase transition is thought to be analogous to the formation of LAT signaling clusters in T cells (*Ganti et al., 2020*). Using this system, we checked whether Btk could be recruited into the reconstituted LAT clusters through its interaction with Grb2.

Single molecules of Btk-Cy5 were tracked and used to compile step-size distributions that reflect the diffusive behavior of Btk under a given condition: the shorter the step-sizes, the slower moving the Btk molecules (*Lin et al., 2020*). When Btk was incubated with phosphorylated LAT alone the step-size distribution shows a fast diffusing population, similar to that observed for Btk alone (*Figure 5B*). Addition of Grb2 shifts the step-size distribution to shorter steps, suggesting two possible situations. One possibility is that Grb2 is able to simultaneously bind Btk and LAT, thus slowing Btk molecules through an additional anchor point to the membrane (via the Grb2-LAT complex). The second possibility is that Grb2 alone has promoted small LAT condensates that are not immediately visible by eye (*Lin et al., 2021*), creating small domains of dense LAT, within which Btk cannot diffuse freely. The addition of SOS-PRR along with Grb2 induces the full phase transition of the phosphorylated LAT (*Figure 5C* and *Figure 5—figure supplement 1*). The step-size distribution shifts even further left under this condition, suggesting that Btk has been trapped within the LAT dense phase (*Figure 5B*). We found that Btk was not able to bind SOS-PRR directly (*Figure 5—figure supplement 1*). These observations suggest that Btk is likely to be tethered to the phosphorylated LAT molecules through binding of the SH2 domain of Grb2 to phosphotyrosine residues on LAT and binding of the SH3 domains of Grb2 to Btk. This interaction leads to recruitment of Btk into the LAT condensate.

## Ideas and speculation

In this paper we present the discovery of an unexpected role for the scaffold protein Grb2 in the control of Btk activity. We show that Grb2 can bind to and enhance the kinase activity of Btk in the presence of PIP$_3$. Previous studies have shown that the N-terminal SH3 domain of Grb2 could bind to Btk through interaction with mIgG tails and SLP65 and thereby potentiate downstream signaling (*Engels et al., 2014*; *Kurosaki and Tsukada, 2000*). Here we show that detectable activation of Btk by Grb2 only occurs when the PH-TH module of Btk engages PIP$_3$ at the membrane. Thus, Grb2 activation of Btk is layered upon a necessary first step of PIP$_3$ generation, which requires BCR stimulation. Enhanced activation of Btk results in increased phosphorylation of Btk itself as well as phosphorylation of the PLCγ2-peptide fusion.

One interesting aspect of our findings is the apparent need for Grb2 dimerization, without any change in the dimerization propensity of Btk upon Grb2 binding. The idea that Grb2 dimers are necessary comes from the observation that the Y160E mutation, a mutation thought to abolish Grb2 dimers, results in much less enhancement of Btk catalytic activity. Additionally, the requirement for all Grb2 domains to be present for full enhancement of catalytic activity of Btk also supports this idea (*Figure 4A*). The crystal structure of Grb2 (PDB code 1GRI) shows the formation of a Grb2 dimer in which there are extensive interactions between the SH3 domains of one Grb2 molecule and the SH2 domain of the other (*Maignan et al., 1995*). Deletion of the SH2 domain would disrupt the dimeric arrangement seen in the crystal structure. One straightforward way that Grb2 dimers could promote Btk activity is through promoting the formation of Btk dimers, thereby promoting trans-autophosphorylation of Btk. Nevertheless, we observe that the population of Btk dimers on PIP$_3$-containing bilayers remains the same in the presence of Grb2 (*Figure 4B*). It is possible that our

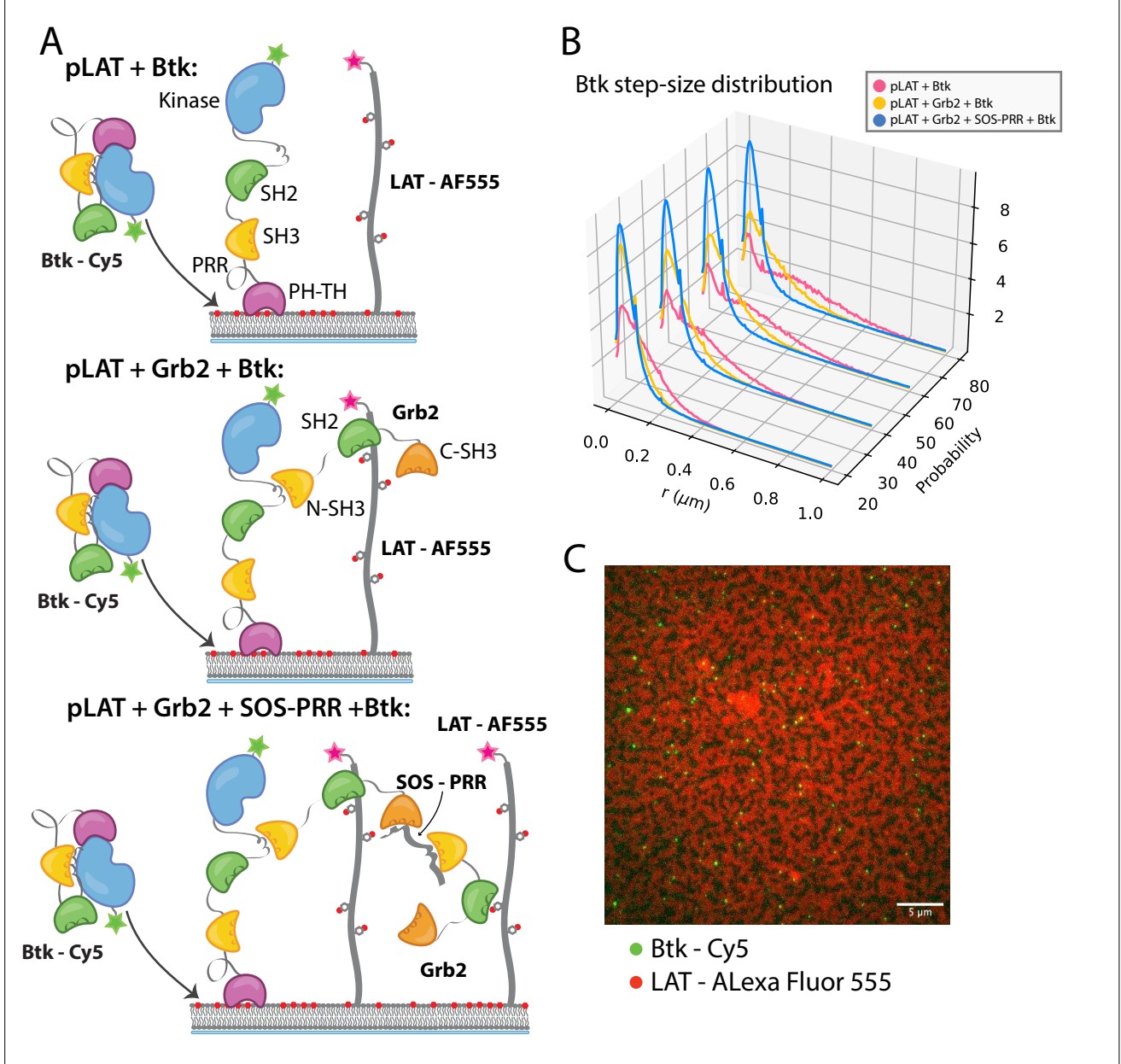

**Figure 5.** Btk can be recruited to scaffold proteins through interaction with Grb2. (**A**) Cartoon schematic of predicted mechanism for Btk recruitment into LAT signaling clusters. (**B**) Step-size distribution for Btk-Cy5 under each condition: phospho-LAT, phospho-LAT + Grb2, phase transitioned LAT (phospho-LAT + Grb2 + SOS-PRR). The step-size distribution was calculated at multiple delay times. These delay times represent frame skips that were taken to ensure that no artifacts arise from selection of the frame rate or size of the camera pixels relative to step-size of the molecules. For each delay time, the dataset used to create the step-size distribution consists of frames from every 20 ms (every frame), 40 ms (every other frame), 60 ms, or 80 ms (every fourth frame). Six different positions across the bilayer were recorded for 500–600 frames each, and one independent experiment was used to confirm trends observed here. (**C**) Overlay of image of Btk-Cy5 (green) and LAT-Alexa Fluor 555 (red) after LAT phase transition. See Dryad repository for source data for this figure, Figure 5—source data 1 (https://dx.doi.org/10.5061/dryad.prr4xgxrf).

The online version of this article includes the following video and figure supplement(s) for figure 5:

**Figure supplement 1.** Behavior of Btk on PIP3-containing membranes with the addition of SOS proline rich region.

**Figure 5—video 1.** Btk does not diffuse freely within phases that are dense in LAT.

https://elifesciences.org/articles/82676/figures#fig5video1

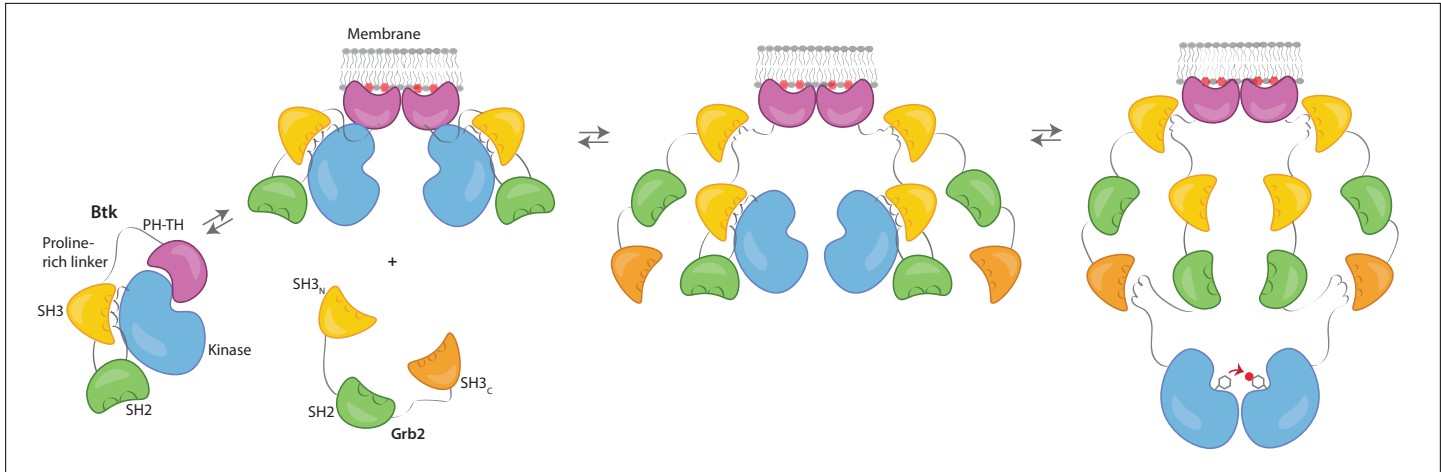

**Figure 6.** Grb2 enhances Btk activation at the membrane. Our data are consistent with a model in which PIP₃ binding at the membrane is not sufficient for full activation, and Btk is still able to maintain some autoinhibition after membrane recruitment. Upon recruitment of Grb2 to the proline-rich linker of Btk, the second Grb2 SH3 domain is able to bind the SH2-kinase linker of Btk and displace Btk's SH3 domain, resulting in full release of autoinhibition. Here we are showing Grb2 as a monomer, however, as discussed, it is possible that Grb2 is binding Btk as a dimer as well.

diffusion measurements are not sensitive enough to detect Grb2 enhancement of Btk dimerization, and further study is required to understand the role of Grb2 dimerization on the activation of Btk.

We propose that Grb2 could activate Btk by first binding the proline-rich linker of Btk and subsequently displacing the Btk SH3 domain through binding of the SH2-Kinase linker, then releasing the inhibitory contacts within the Src module of Btk (*Figure 6*). This speculation is based on how Src-family kinases are activated by binding of the SH2-kinase linker by HIV Nef protein (*Aryal et al., 2022*; *Moarefi et al., 1997*). We demonstrate that Btk kinase activity increases with the addition of Grb2 only with simultaneous availability of PIP₃, which is an important point because Grb2 is a ubiquitous protein that is highly expressed in many cells (*Shi et al., 2016*). This ensures that the Btk signal remains responsive to activation of the B-cell receptor.

By looking directly at Btk phosphorylation in the presence of Grb2 in our reconstituted system, we reveal an additional consequence of Grb2 binding – Grb2 binding can recruit Btk to signaling clusters at the membrane. In particular, the increased phosphorylation by Btk that we observe for a segment of PLCγ2 in the presence of Grb2 shows how Grb2 binding of Btk could have a direct impact on the downstream signaling of Btk, subsequently increasing the population of active PLCγ2 molecules at the membrane. In the absence of Grb2, Btk activation is slow, even when PIP₃ levels are high and can promote dimerization. In the presence of Grb2, Btk phosphorylation proceeds much more rapidly. This work illuminates a new level of regulation within Btk, in which optimal signaling may rely on interaction with an adaptor molecule that both stimulates activity and facilitates localization with downstream substrates.

## Materials and methods

### Key resources table

| Reagent type (species) or resource | Designation | Source or reference | Identifiers | Additional information |
|---|---|---|---|---|
| Recombinant DNA reagent | His₆-TEV (Tobacco etch virus) site-Grb2 pET28 (plasmid) | This paper | | Full-length, UniProt P62993 |
| Recombinant DNA reagent | Btk pET28 (plasmid) | This paper | | Full-length, UniProt Q06187 |
| Recombinant DNA reagent | Btk SH3-SH2-kinase pET28 (plasmid) | This paper | | Residues 212–659 |
| Recombinant DNA reagent | Btk SH2-Kinase pET28 (plasmid) | This paper | | Residues 281–659 |

*Continued on next page*

*Continued*

| Reagent type (species) or resource | Designation | Source or reference | Identifiers | Additional information |
|---|---|---|---|---|
| Recombinant DNA reagent | Btk kinase pET28 (plasmid) | This paper | | Residues 402–659 |
| Recombinant DNA reagent | Btk proline-rich linker pET28 (plasmid) | This paper | | Residues 171–214 |
| Recombinant DNA reagent | PLCγ2-peptide fusion pET28 (plasmid) | This paper | | PLCγ2 Residues 746–766 UniProt P16885, EGFP |
| Recombinant DNA reagent | His$_6$-TEV site-Grb2 SH3$_N$-SH2 pET28 (plasmid) | This paper | | Residues 1–153 |
| Recombinant DNA reagent | His$_6$-TEV site-Grb2 SH2-SH3$_C$ pET28 (plasmid) | This paper | | Residues 60–217 |
| Recombinant DNA reagent | His$_6$-TEV site-Grb2 SH3$_N$-SH3$_C$ pET28 (plasmid) | This paper | | Residues 1–59, 154–217 |
| Recombinant DNA reagent | His$_6$-TEV site-Grb2 SH3$_N$ pET28 (plasmid) | This paper | | Residues 1–59 |
| Recombinant DNA reagent | His$_6$-TEV site-Grb2 SH2 pET28 (plasmid) | This paper | | Residues 60–153 |
| Recombinant DNA reagent | His$_6$-TEV site-Grb2 SH3$_C$ pET28 (plasmid) | This paper | | Residues 154–217 |
| Recombinant DNA reagent | His$_6$-TEV site-Grb2 R86K pET28 (plasmid) | This paper | | Full-length, R86K |
| Recombinant DNA reagent | His$_6$-TEV site-Grb2 Y160E pET28 (plasmid) | This paper | | Full-length, Y160E |
| Recombinant DNA reagent | His$_6$-SUMO-Btk T403AzF pET28 (plasmid) | This paper | | Full-length, T403Amber STOP |
| Recombinant DNA reagent | His$_6$-TEV site-LAT pET28 (plasmid) | This paper | | Residues 252–520, UniProt P08631 |
| Recombinant DNA reagent | His$_6$-TEV site-Hck pET28 (plasmid) | This paper | | Residues 1051–1333, UniProt Q07889 |
| Recombinant DNA reagent | His$_6$-TEV site-SOS-PRR pET28 (plasmid) | This paper | | UniProt P15273 |
| Recombinant DNA reagent | YopH pCDFDuet (plasmid) | This paper | | |
| Strain, strain background (*Escherichia coli*) | BL21 (DE3) | Thermo Fisher | C600003 | Chemically competent cells |
| antibody | Anti-Phospho-Tyrosine mAb, pTyr-100 (Mouse monoclonal) | Cell Signaling | Cat#: 9411 | WB (1:2000) |
| antibody | Anti-Mouse HRP linked antibody (Horse) | Cell Signaling | Cat#: 7076 | WB (1:5000) |
| Other | 4-Azido-L-phenylalanine HCl | Amatek Chemical Co. | Cat#: 33173-53-4 | |
| Other | dibenzocyclooctyl (DBCO)-Cy5 | Click Chemistry Tools | Cat#: A130 | |
| Other | AlexFluor 647 C$_2$ Maleimide | Thermo Fisher, | Cat#: A20347 | |
| Other | AlexFluor 555 C$_2$ Maleimide | Thermo Fisher | Cat#: A20346 | |
| Other | PI(3,4,5)P3 di16 (ammonium salt) | Echelon Biosciences, Inc | Cat#: P-3916A | |
| Other | 1,2-dioleoyl-sn-glycero-3-[(N-(5-amino-1-carboxypentyl)iminodiacetic acid)succinyl] (nickel salt) (DGS-NTA(Ni)) | Avanti Polar Lipids | Cat#: 790404 | |

*Continued*

| Reagent type (species) or resource | Designation | Source or reference | Identifiers | Additional information |
|---|---|---|---|---|
| Other | 18:1 1,2-dioleoyl-sin-glycero-3-phosphocholine (DOPC) | Avanti Polar Lipids | Cat#: 850375 | |
| Software, algorithm | Trackmate | FIJI (Image J) | Version: 2.9.0/1.53t | |
| Software, algorithm | Imtrack package | Groves lab (Darren McAffee, PhD) | https://github.com/lnocka/Single_molecule_tracking.git | |
| Software, algorithm | multi_chi_global.py | This paper | https://github.com/lnocka/Single_molecule_tracking.git | |
| Software, algorithm | analysis_overlay_3D_line.py | This paper | https://github.com/lnocka/Single_molecule_tracking.git | |

## Protein preparation

For preparation of full-length Btk, the plasmid was transformed into BL21(DE3) *Escherichia coli* (*E. coli*) containing the YopH expression plasmid (described above) and plated on kanamycin and strep-tomycin containing agar plates (*Seeliger et al., 2005*). Transformed cells were first grown in a 200 mL Terrific broth containing 100 µg/mL kanamycin and streptomycin overnight culture at 37 °C. This was then split into 6 L of Terrific broth containing 100 µg/mL kanamycin and streptomycin and grown to an optical density of 1–1.5 at 37 °C. The cultures were mixed 1:1 with new media and antibiotic at 4 °C and 1 mM isopropyl β-D-1-thiogalactopyranoside (IPTG) and maintained at 4 °C to grow overnight. After overnight expression, the cultures were spun down and resuspended in 15–30 mL of Ni A buffer (500 mM NaCl, 20 mM Tris-HCl pH 8.5, 20 mM Imidazole, 5% Glycerol). These were then flash frozen and kept at –80 °C until the next step of the purification.

Cell pellets were thawed and then lysed by homogenization or sonication with addition of phenyl-methylsulfonyl fluoride. The lysate was then spun down at 16,500 rpm for 1 hr. Supernatant was collected and flowed over a HisTrap FF (Cytiva, product no. 17525501) column. The column was then washed with 10 column volumes (CV) Ni A buffer, followed by four CV Ni B buffer (500 mM NaCl, 20 mM Tris-HCl pH 8.5, 500 mM Imidazole, 5% Glycerol). The elution fraction was then loaded onto a desalting column equilibrated in Buffer A (150 mM NaCl, 20 mM Tris-HCl pH 8.0, 5% Glycerol). The protein peak was collected and incubated with His-tagged ULP1 protease overnight at 4 °C.

After SUMO cleavage, the sample was run over a second HisTrap column, and this time the flow through and wash were collected. In some cases, a gradient of imidazole was applied as the Btk PH domain has some affinity for the Ni column even in the absence of the His tag. The protein was then concentrated in an Amicon Ultra-15 centrifugal filter 10 kDa or 30 kDa molecular weight cutoff depending on the sample (Millipore Sigma, product no. UFC9010 or UFC9030, respectively) to less than 2 mL total volume. This was then loaded onto an HiLoad 16/60 Superdex 200 (Cytiva) column equilibrated in Buffer A for biochemistry or Buffer B (150 mM NaCl, 50 mM Hepes pH 7.4, 5% Glyc-erol) for imaging studies. Fractions containing the purest target protein were collected and concen-trated. These were then aliquoted and stored at –80 °C after flash freezing.

All other constructs were purified in a similar manner with the following changes. All constructs that did not contain a kinase domain were transformed into BL21(DE3) *E. coli* without the YopH plasmid, and therefore they were grown with only kanamycin. Expression for these constructs was carried out overnight at 18 °C. Grb2 constructs and PLCγ2-peptide fusion express at much higher levels than the Btk constructs and therefore only 1–2 L were prepared at a given time, and therefore no overnight culture was required prior to protein expression. For all constructs that contain a His tag after purifi-cation (LAT, Hck, and His-tagged Btk constructs), the protease and second HisTrap column were elim-inated. Grb2 constructs and SOS-PRR each contain a TEV site and therefore His tagged TEV protease was used to remove the His tag.

## Protein fluorescent labeling by maleimide conjugation

Grb2 and SOS-PRR were each prepared at a concentration of 50 µM and incubated with 5 mM DTT for 30 min on ice to ensure all accessible cysteines were reduced. AlexaFluor 647 $C_2$ Maleimide (Thermo

Fisher, A20347) or AlexaFluor 555 $C_2$ Maleimide (Thermo Fisher, A20346) was dissolved in anhydrous DMSO and added in equimolar amounts to Grb2 or threefold molar excess to SOS-PRR and incubated at room temperature (approximately 22 °C) for 30 min or at 4 °C overnight, depending on the stability of the protein. The reaction was then quenched with 10 mM DTT for 30 minutes at 4 °C. After quenching, the protein was diluted in 10–15 mL fresh buffer, and then concentrated in an Amicon Ultra-15 centrifugal filter (Millipore Sigma), allowing free dye to be removed in the flow through. This process was continued until no more free dye could be easily detected in the flow through. At this point the protein was then purified by gel filtration in Buffer B as described above. Labeling efficiency was calculated based on the absorbances at 280 nm and the peak excitation wavelength for the dye, taking into account dye contribution at 280 nm. A note about Grb2 labeling: the protein tends to aggregate when exposed to an excess of dye, and therefore we favored under labeling by only providing equimolar amount of dye.

## Protein fluorescent labeling by azido phenylalanine incorporation

For site specific labeling of full-length Btk for single-molecule studies, we used unnatural amino acid incorporation of an Azido phenylalanine (AzF) residue (*Amiram et al., 2015*; *Bard and Martin, 2018*; *Chatterjee et al., 2013*;; *Chin et al., 2002*). An amber codon (UAG) was introduced at the desired labeling position in the plasmid for Btk expression, and it was ensured that the stop codon for this gene was not amber. This plasmid was co-transformed into BL21(DE3) containing a plasmid expressing YopH with the pUltra-pAzFRS plasmid. A 5 mL starter culture was grown overnight in TB at 37 °C from this transformation. This was used to inoculate 1 L TB the following morning which was grown to an optical density between 0.5 and 1 at 37 °C. This culture was spun down and resuspended in 167 mL cold (4 °C) TB containing 2 mM 4-Azido-L-phenylalanine HCl (Amatek Chemical Co., CAS no. 33173-53-4) before induction, this culture was allowed to grow for 1.25 hr at 4 °C. The culture was then induced with 1 mM IPTG and grown overnight at 4 °C. After this, protein purification proceeded as normal until just before the gel filtration step, except that all reducing agents were left out of the buffers. Yield of the protein is drastically reduced, however for our applications a low yield was not a problem as long as labeling was feasible.

For labeling, the purified protein was concentrated to about 300 μL. 5 mM 5,5-dithio-bis-(2-nitrobenzoic acid) (DTNB, Ellman's reagent) was prepared in 50 mM HEPES, pH 7.0, 250 mM KCl, 5% glycerol. DTNB was added at 15 X molar excess to the protein solution and incubated at room temperature for 10 min. The protein was then cooled back to 4 °C and 300 μM dibenzocyclooctyl (DBCO)-Cy5 (Click Chemistry Tools, Catalog no. A130) was added from a 30 mM stock in DMSO. This was incubated overnight at 4 °C. The reaction was quenched with 5 mM DTT and incubated at 4 °C for 1 hr. At this point the labeled protein was purified by gel filtration, then concentrated to an appropriate concentration, flash frozen and stored at –80 °C. This protocol was adapted from the lab of Andreas Martin (*Bard et al., 2019*; *Bard and Martin, 2018*; *Lander et al., 2012*).

## Western blot assays for Btk kinase activity

Samples were prepared with 2 X concentration of each component depending on the condition to be tested (2 μM Btk, 250 μM 4% $PIP_3$ single unilamellar vesicles (SUVs), 0–20 μM Grb2) in Buffer A, final concentrations during the reaction are indicated for each blot. These components were incubated for 15 min at room temperature and then diluted 1:1 in activation buffer (20 mM $MgCl_2$, 2 mM ATP, 2 mM sodium vanadate). These were mixed and the reaction proceeded at room temperature for the designated amount of time, indicated for each blot separately. The reaction was then quenched by mixing 1:1 with Quench buffer (166 mM Tris-HCl pH 6.8, 10% SDS, 10 mM DTT, 3 μM bromophenol blue, 10% glycerol, 100 mM EDTA). These samples were then immediately heated at 90 °C for 15 min, followed by loading onto two 12 or 15% SDS page gels, which were run at 250 V for 35 min. One gel was stained with Coomassie blue, the second was prepared for a western blot transfer.

Filters were soaked in western blot transfer buffer (25 mM Tris-HCl pH 7.4, 192 mM glycine) supplemented with 0.1% SDS. The membrane was activated in MeOH for 1 min and then transferred to western blot transfer buffer supplemented with 20% MeOH. The protein was transferred in a semi-dry apparatus to the membrane at 25 V for 1 hr. This membrane was then blocked in 5% Carnation non-fat milk powder for one hour while shaking gently. To probe for phosphotyrosine, the membrane was then transferred to a 1:2000 dilution of Phospho-Tyrosine Mouse mAb (pTyr-100) (Cell Signaling,

catalog no. 9411) for shaking overnight at 4 °C. The following day, the membrane was washed four times for 10 min each in TBS-T (20 mM Tris pH 7.5, 150 mM NaCl, 0.1% Tween-20). The membrane was then transferred to a secondary antibody solution containing a 1:5000 dilution of Anti-Mouse HRP linked antibody (Cell Signaling, catalog no. 7076) for 1 hr of shaking at room temperature. The membrane was then washed again four times for 10 min each in TBS-T. At this point the blot was imaged with Western Bright ECL HRP Substrate (Advansta, catalog no. K-12045-D20).

Blots were quantified using Fiji (ImageJ). Each lane of the gel was selected with a rectangle and Fiji Analyze Gels option. The lanes were then plotted and a linear baseline was marked. Any peaks in close proximity were separated with a vertical line. The wand tool was then used to select the peak area. These intensities were then plotted. In the case where a 0 timepoint was recorded, or if a titration of Grb2 was being assessed and a point with no Grb2 was added was recorded, these were used as baseline to define a change in intensity for the remaining samples (*Figure 3B–D*). It should be noted that this technique is only semi-quantitative as the intensity may not be linear with phospho-tyrosine concentration; these graphs are provided as a tool to compare trends rather than exact values. Replicates are provided to ensure that the observed trend was reproducible.

## Supported-lipid bilayer preparation

Vesicles were prepared by first mixing the desired ratios of $PIP_3$ (Echelon Biosciences, Inc), 1,2-dioleoyl-sn-glycero-3-[(N-(5-amino-1-carboxypentyl)iminodiacetic acid)succinyl] (nickel salt) (DGS-NTA(Ni)), and 18:1 1,2-dioleoyl-sin-glycero-3-phosphocholine (DOPC) (Avanti Polar Lipids). For experiments with full-length Btk (no His tag) 4% $PIP_3$, 96% DOPC mixtures were prepared. For experiments using His-tagged Btk constructs 4% DGS-NTA(Ni), 96% DOPC mixtures were prepared. And for experiments involving LAT 4% $PIP_3$, 4% DGS-NTA(Ni), 92% DOPC bilayers mixtures were prepared. All lipids were stored in solution with chloroform, except $PIP_3$ which was stored in powder form and dissolved in a solution of 1:2:0.8 Chloroform:Methanol:Water just before use. After mixing, the solution was dried in an etched 50 mL round bottomed flask by rotary evaporation for 15 min and then under nitrogen for 15 min. Dried lipids could then be kept overnight at 4 °C, sealed from air or used immediately.

The lipids were rehydrated in water to 1 mg/mL lipid concentration by vortexing. Single unilamellar vesicles (SUVs) were created by sonication (Analis Ultrasonic Processor 750 watt, 20 kHz) of the lipids at 33% power, 20 s on, 50 s off for 1 min 40 s of active sonication. A solution of 0.25 mg/mL SUVs in 0.5 X TBS was prepared. These were added to a Sticky Slide VI 0.4 (Ibidi GMBH) attached to a Piranha-etched glass slide, 80 µL of SUVs per well. The SUVs were incubated at room temperature for 30 min. Each well was then washed with 1 mL of Hepes Buffered Saline (HBS) under vacuum, being careful to avoid introduction of air bubbles to the chamber. When using DGS-NTA(Ni), the wells were then incubated with 100 mM $NiCl_2$ for 5 min, and again washed with 1 mL HBS. At this point, the wells were blocked with either 1 µg/mL Poly(L-Lysine)-PEG (PLL-PEG) for 1 min, or 1 mg/mL Bovine Serum Albumin (BSA) or β-Casein for 10 min. The blocking agent for each experiment was optimized by checking the mobile fraction of surface-bound particles. In the case of full-length Btk on 4% $PIP_3$ bilayers, 1 µg/mL PLL-PEG yielded the best results. For LAT containing bilayers, 1 mg/mL BSA was used to block. For all other experiments, 1 mg/mL β-Casein yielded the best results.

For experiments where DGS-NTA(Ni) lipids were used to coordinate His-tagged protein to the bilayers, the protein was added at the desired incubation concentration and left at room temperature for 40 min. The chambers were then washed with 600 µL HBS gently by hand and incubated for 20 more minutes at room temperature. They were then again washed with 600 µL HBS. Finally, they were washed with 100 µL imaging buffer containing 100 µg/mL BSA or β-Casein (depending on the blocking agent that was used initially) and 10 mM BME in HBS.

## Microscopy

Total internal reflection fluorescence adsorption experiments were carried out and collected on a Nikon Eclipse Ti-inverted microscope (Nikon, Tokyo, Japan) with 100 x objective and Andor iXon electron-multiplying charge-coupled device (EMCCD) camera (Oxford Instruments), as previously described (*Bhattacharyya et al., 2020*). Adsorption curves were acquired with 15 second intervals with a laser power of 0.5–1 mW and exposure time of 75 ms, and data are displayed as a difference intensity, where the baseline for a given sample was calculated from the average of the four frames preceding Grb2 addition.

Single-molecule traces were recorded as described previously (*Lin et al., 2020*). Btk was allowed to equilibrate with the supported-lipid bilayers for 30 min before imaging. This was determined to be sufficient for Btk to equilibrate on the bilayers based on previous work with the PH-TH module alone (*Chung et al., 2019*). Movies were recorded at exposure time of 20 ms, magnification of ×1.5 on ×100 objective, and laser power of 20 mW. Five or more traces of 500–600 frames were collected at various places across the sample.

Fluorescent molecules in these movies were tracked using the TrackMate plugin from Fiji (Image J) (*Tinevez et al., 2017*). Particles were identified using the difference of Gaussian detector. Tracking parameters were kept consistent across experiments, 0.5 µm diameter spot size, threshold of 100 (determined by visual inspection), particle links determined using simple linear assignment problem tracker, with linking distance maximum of 1.5 µm, a maximum frame gap of 2, and maximum gap distance of 1.5 µm. An immobile fraction of fluorescent Btk was always observed, and this fraction was included in the analysis and showed minimal change across samples. Tracks were analyzed by calculating a step-size distribution for all tracked particles. In order to ensure that step size was sufficiently long compared to the camera pixel size, we calculated the step-size distribution for four different time delays (every 2 frames, 3 frames, 4 frames, and 5 frames). To prevent over-counting, these data were drawn from skips within the original dataset. The full data-set, including all time delays, was then fit to a three-component diffusion model:

$$\rho\left(r, \tau, D\right) = \sum_{i=0}^{3} \alpha_i \frac{r}{2D\tau} e^{\frac{-r^2}{4D\tau}}$$

Where $r$ is molecular displacement, $\tau$ represents the delay time, $D$ is the diffusion coefficient, and $\alpha_i$ is the contribution from each component. The fastest diffusion coefficient is reported, as the other two remain consistent across all Btk concentrations. Error reported is the standard deviation determined from the fit distribution. Examples of the step-size distributions and their corresponding fits can be found in *Figure 4—figure supplement 3*.

## Sample preparation for liquid chromatography-mass spectrometry

Two 20 µL reactions were prepared with His6-SUMO-Plcg2 peptide (35 µM), Grb2 (3.5 µM), 4% PIP$_3$ lipid vesicles (250 µM), tris-buffered saline (0.25%), magnesium chloride (10 mM), ATP (1 mM), and sodium vanadate (1 mM), using gel-filtration buffer (25 mM Tris, pH 8.0, 150 mM sodium chloride, 1 mM TCEP, and 5% glycerol) for all dilutions. Btk (3.5 µM) was added to one of the two reactions and both aliquots were incubated at room temperature for 1 hr. Reactions were quenched by addition of urea (4 M) and DTT (6.2 mM) followed by a 55 °C incubation for 20 min. Samples were then alkylated with iodoacetamide (12.9 mM) for 30 min in the dark at room temperature. Following this incubation, samples were digested overnight at room temperature in a buffer containing tris (50 mM, pH 8.0), calcium chloride (1 mM), and mass-spectrometry grade trypsin (0.01 mg/mL, Trypsin/Lys-C Mix, Mass Spec Grade, Promega, catalog no. V5071).

## Liquid chromatography-mass spectrometry

Samples of trypsin-digested proteins were analyzed using a liquid chromatography (LC) system (1200 series, Agilent Technologies, Santa Clara, CA) that was connected in line with an LTQ-Orbitrap-XL mass spectrometer equipped with an electrospray ionization (ESI) source (Thermo Fisher Scientific, Waltham, MA). The LC system contained the following modules: G1322A solvent degasser, G1311A quaternary pump, G1316A thermostatted column compartment, and G1329A autosampler unit (Agilent). The LC column compartment was equipped with a Zorbax 300 SB-C8 column (length: 150 mm, inner diameter: 1.0 mm, particle size: 3.5 µm, part number: 863630–906, Agilent). Acetonitrile, formic acid (Optima LC-MS grade, 99.5+%, Fisher, Pittsburgh, PA), and water purified to a resistivity of 18.2 MΩ·cm (at 25 °C) using a Milli-Q Gradient ultrapure water purification system (Millipore, Billerica, MA) were used to prepare LC mobile phase solvents. Solvent A was 99.9% water/0.1% formic acid and solvent B was 99.9% acetonitrile/0.1% formic acid (volume/volume). The elution program consisted of isocratic flow at 1% (volume/volume) B for 2 min, a linear gradient to 35% B over 30 min, a linear gradient to 95% B over 1 min, isocratic flow at 95% B for 5 min, a linear gradient to 1% B over 1 min, and isocratic flow at 1% B for 21 min, at a flow rate of 120 µL/min. The column compartment was maintained at 40 °C and the sample injection volume was 10 µL. External mass calibration was

performed in the positive ion mode using the Pierce LTQ ESI positive ion calibration solution (catalog number 88322, Thermo Fisher Scientific) prior to running samples. Full-scan, high-resolution mass spectra were acquired in the positive ion mode over the range of mass-to-charge ratio ($m/z$)=340–1800 using the Orbitrap mass analyzer, in profile format, with a mass resolution setting of 60,000 (at $m/z$=400, measured at full width at half-maximum peak height). In the data-dependent mode, the ten most intense ions exceeding an intensity threshold of 10,000 raw ion counts were selected from each full-scan mass spectrum for tandem mass spectrometry (MS/MS) analysis using collision-induced dissociation (CID). MS/MS spectra were acquired using the linear ion trap, in centroid format, with the following parameters: isolation width 3 $m/z$ units, normalized collision energy 28%, default charge state 3, activation Q 0.25, and activation time 30 ms. Real-time charge state screening was enabled to exclude unassigned charge states from MS/MS analysis. To avoid the occurrence of redundant MS/MS measurements, real-time dynamic exclusion was enabled to preclude re-selection of previously analyzed precursor ions, with the following parameters: repeat count 2, repeat duration 10 s, exclusion list size 500, exclusion duration 60 s, and exclusion mass width ±10 parts per million. Data acquisition was controlled using Xcalibur software (version 2.0.7, Thermo Fisher Scientific). Raw data files were searched against the amino acid sequences of the Btk, Grb2, and His$_6$-Sumo-eGFP-Plcg2 proteins using Proteome Discoverer software (version 1.3, SEQUEST algorithm, Thermo Fisher Scientific), for tryptic peptides (i.e. peptides resulting from cleavage C-terminal to arginine and lysine residues, not N-terminal to proline residues) with up to two missed cleavages and carbamidomethylcysteine, dehydrocysteine (i.e. cystine), methionine sulfoxide and phosphotyrosine as dynamic post-translational modifications. Assignments were validated by manual inspection of MS/MS spectra.

## Reconstitution of LAT phase preparation on supported-lipid bilayers

LAT was reconstituted on supported-lipid bilayers as described (*Huang et al., 2017b*). Supported-lipid bilayers were prepared as described above with 30 nM His$_6$-Hck and 150 nM His$_6$-LAT-Alexa Fluor 555 on a 4% DGS-NTA(Ni), 96% DOPC. The LAT was phosphorylated by including 1 mM ATP and 10 mM MgCl$_2$ in the imaging buffer and incubating for 20 min before adding other components. The components of the LAT signaling cluster were added sequentially along with 1 nM Btk T403AzF-Cy5 (Btk-Cy5). In one condition, Btk-Cy5 was added alone to the phosphorylated LAT containing bilayers, in another condition 5.8 µM Grb2 and 1 nM Btk-Cy5 were added together, and in a third condition 5.8 µM Grb2, 1.45 µM SOS-PRR and 1 nM Btk-Cy5 were all added together (*Figure 5A*). For the final condition, the bilayers were allowed to incubate for 1 hr to promote formation of the condensed-phase LAT domains. We note that fluorescently labeled SOS-PRR cannot be recruited to supported-lipid bilayers containing Btk alone (*Figure 5—figure supplement 1*). Multiple traces of Btk diffusion were recorded and analyzed as described above for single molecule tracking.

## Acknowledgements

We thank Jean Chung, William Huang, and Josh Cofsky for their help in providing protocols and troubleshooting assistance. We thank Darren McAffee and Kiera Wilhelm for sharing their data analysis pipelines for the single molecule studies, and Chun-Wei Lin, Joey DeGrandchamp, and Nugent Lew for sharing protein that was used for some experiments in this paper. We are grateful to Neel Shah, Jeanine Amacher, and Helen Hobbs for helpful conversations about kinase signaling. Finally, we thank Susan Marqusee and David Wemmer for their feedback on this project. TJE is a Damon Runyon Fellow supported by the Damon Runyon Cancer Research Foundation (DRG-2429–21). The QB3/Chemistry Mass Spectrometry Facility received National Institutes of Health support (grant number 1S10OD020062-01). This work was supported by the National Institutes of Health (grant number PO1 A1091580)

## Additional information

### Competing interests

John Kuriyan: Co-founder of Nurix Therapeutics. The other authors declare that no competing interests exist.

## Funding

| Funder | Grant reference number | Author |
|---|---|---|
| Damon Runyon Cancer Research Foundation | DRG-2429-21 | Timothy J Eisen |
| National Institutes of Health | 1S10OD020062-01 | Anthony T Iavarone |
| National Institutes of Health | PO1 A1091580 | Jay T Groves |

The funders had no role in study design, data collection and interpretation, or the decision to submit the work for publication.

## Author contributions

Laura M Nocka, Timothy J Eisen, Conceptualization, Data curation, Formal analysis, Validation, Investigation, Visualization, Methodology, Writing – original draft, Writing – review and editing; Anthony T Iavarone, Data curation, Formal analysis, Writing – original draft; Jay T Groves, John Kuriyan, Conceptualization, Supervision, Funding acquisition, Writing – original draft, Writing – review and editing

## Author ORCIDs

Laura M Nocka (iD) http://orcid.org/0000-0003-2556-4227
Timothy J Eisen (iD) http://orcid.org/0000-0002-4812-4171
Jay T Groves (iD) http://orcid.org/0000-0002-3037-5220
John Kuriyan (iD) http://orcid.org/0000-0002-4414-5477

## Decision letter and Author response

Decision letter https://doi.org/10.7554/eLife.82676.sa1
Author response https://doi.org/10.7554/eLife.82676.sa2

# Additional files

## Supplementary files

• MDAR checklist

## Data availability

All data collected and analyzed for this study are included in the manuscript and supporting files or available on Dryad. Custom scripts for single molecule analysis can be found on GitHub (https://github.com/lnocka/Single_molecule_tracking copy archived at *Nocka, 2023*).

The following datasets were generated:

| Author(s) | Year | Dataset title | Dataset URL | Database and Identifier |
|---|---|---|---|---|
| Nocka LM, Eisen T, Iavarone A, Groves JT, Groves JT, Kuriyan J | 2023 | Stimulation of the catalytic activity of the tyrosine kinase Btk by the adaptor protein Grb2: Part 3 | doi:10.5061/dryad.prr4xgxrf | Dryad Digital Repository, 10.5061/dryad.prr4xgxrf |
| Nocka LM, Eisen T, Iavarone A, Groves JT, Groves JT, Kuriyan J | 2023 | Stimulation of the catalytic activity of the tyrosine kinase Btk by the adaptor protein Grb2: Part 2 | doi:10.5061/dryad.jwstqjqfd | Dryad Digital Repository, 10.5061/dryad.jwstqjqfd |
| Nocka LM, Eisen T, Iavarone A, Groves JT, Groves JT, Kuriyan J | 2023 | Stimulation of the catalytic activity of the tyrosine kinase Btk by the adaptor protein Grb2: Part 1 | doi:10.5061/dryad.8sf7m0ctd | Dryad Digital Repository, 10.5061/dryad.8sf7m0ctd |

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
