## [Editor Report]

This important study reports an unexpected mode of activation of the critical immune cell kinase, Btk, by the SH3-SH2 domain-containing adaptor protein Grb2. The authors convincingly demonstrate that Grb2 binding to a Pro-rich region positioned between the Btk PH-TH domains and Src module plays a crucial role in relieving Btk autoinhibition and mediating Btk recruitment into signaling clusters on artificial membranes. These findings provide a mechanistic basis for the clusters previously reported in cells, and resolve in part why Btk can be activated via modes distinct from its close relatives, such as Itk kinase.

---

## [Decision Letter]

**Decision letter after peer review:**

Thank you for submitting your article "Stimulation of the catalytic activity of the tyrosine kinase Btk by the adaptor protein Grb2" for consideration by *eLife*. Your article has been reviewed by 3 peer reviewers, one of whom is a member of our Board of Reviewing Editors, and the evaluation has been overseen by a Reviewing Editor and Amy Andreotti as the Senior Editor. The reviewers have opted to remain anonymous.

Essential revisions:

The review process revealed weaknesses in the manuscript. While the reviewers appreciate the novelty and potential importance of the findings, there is a strong consensus that the manuscript as submitted must be revised. The data in this paper are not sufficiently robust and the manuscript is not as well written as would be expected of an *eLife* publication. The concern is that, as presented, the authors provide potentially interesting and novel insights, but the data are not interpreted in a manner that yields a definitive advance in our mechanistic understanding of BTK regulation and the precise role of Grb2. In line with the magnitude of the changes sought, we anticipate the reviewers will be further consulted on the revisions.

1) All reviewers highlighted that antibodies towards specific pTyr sites would be most appropriate as a read-out of phosphorylation because the pan-pTyr antibody could detect ambiguous/off-target signals. Repeating with specific pTyr antibodies is essential.

2) The variability between duplicate experiments limits robust interpretation throughout the manuscript; it is essential that further independent repeats are performed to ensure the findings are robust.

3) Figure 6. Further negative control experiments are required: with labeled Btk lacking its PH-TH-PRR domains, immobilised on the membrane with a His tag, as they have done in other experiments; and with the pTyr-binding deficient SH2 domain mutant of Grb2.

4) There are suggestions from all 3 reviewers for how the data presentation and writing of the manuscript might be improved. A number of important points relating to data interpretation should also be considered in revision; the novel findings are of interest but data are currently insufficient to support the authors' claims. Extensive editing of the manuscript is required; in its current form, it would not be acceptable for publication.

5) The authors should resolve the discrepant pTyr data in Figure 3 vs 5 Supp2, and present a more complete picture of the analysis of diffusion constants in Figure 5, including estimations of error.

*Reviewer #2 (Recommendations for the authors):*

Figure 3. The authors need to perform sufficient replicates to confidently argue that Grb2 directly stimulates Btk activity. Whilst the two replicates of Btk autophosphorylation are relatively convincing, the data supporting stimulated substrate phosphorylation are not.

Since substrate phosphorylation should be directly influenced by activation loop autophosphorylation of Btk, these two readouts should be mutually consistent.

Furthermore, an inspection of the change in Btk autophosphorylation in Figure 3 panel C is not consistent with that reported for the same time point (5 min) in panel A, suggesting that the authors suffer from a lack of reproducibility in these experiments.

Figure 4. Again, insufficient replicates are reported to support the authors' claims. In particular, the assertion that Grb2 increases activity of the SH3-SH2-kinase module is based on a single time point (5 min) with two measurements, as the error bars of the other time points overlap between the conditions with and without Grb2. Furthermore, the authors do not observe any significant difference in substrate phosphorylation, which would be an expected consequence of activation loop autophosphorylation.

Figure 5. Missing error bars for diffusion coefficient in panel A (presumably these were the averages of multiple particle tracks?). Again, insufficient replicates for B-C. The conceptual problem here: the authors have designed what they say is a rescue experiment to test whether Grb2 can 'rescue' autophosphorylation of a mutant of Btk impaired in its dimerization on the membrane. However, an inspection of their data reveals no visible defect in Btk autophosphorylation between the wt and mutant proteins (Figure 5 Supplement 2). Furthermore, in order to compare wt with mutant Btk, one would need to know whether the level of Btk phosphorylation detected is sub-stoichiometric or not. Otherwise, Btk may already be maximally phosphorylated with just PIP3-liposomes and an effect of Grb2 would not be detectable. The authors should also comment on why Grb2 is phosphorylated in these experiments, but not in those experiments reported in Figure 3. In the text, the authors reference Figure 3 Supplement 2 as evidence for Grb2 phosphorylation, when in fact there is no sign of Grb2 phosphorylation, and a signal for Grb2 phosphorylation is only found in Figure 5 Supplement 2.

Figure 6. To address the concerns with the interpretation of the data presented in this figure, it is recommended to the authors that they perform a negative control experiment with labeled Btk lacking its PH-TH-PRR domains, immobilised on the membrane with a His tag, as they have done in other experiments. The diffusion properties of Btk SH3-SH2-kinase bound to DGS-NTA(Ni) vs full-length Btk bound to PIP3 should be comparable. This control experiment will address whether the reduction in Btk mobility the authors observe is specific to its Grb2-dependent recruitment into LAT condensates, or, whether the condensates create physical barriers to Btk diffusion on the membrane.

The text is unnecessarily long and repetitive, as well as confusing in places, and contains a number of factual errors. The cartoon schematics in multiple figures are often not representative of the experiment being performed.

*Reviewer #3 (Recommendations for the authors):*

Specifically, I recommend:

1. In the abstract, the authors have included a citation for another study. This is highly unconventional and I recommend removing it.

2. In schematics, the component domains are not always labelled. There's a great deal of assumed knowledge in doing so, but this does not allow the work to be interpreted by a generalist audience, such as *eLife*'s readership.

3. In the introduction, the Saraste dimer is introduced but the orientation/arrangement of domains is not described or depicted in a way that a non-expert would be able to interpret. The dimer is not shown at all in Figure 1 but should be.

4. Line 110. Crosslinking suggests a covalent linkage. I urge the authors to reconsider their choice of language.

5. Greater clarity on labelling methods in the Results and Discussion would aid interpretation both where TIRF is first mentioned and later where Amber UAA methods are used. In both instances, it would be helpful to the reader to state clearly that these studies use recombinant proteins and a synthetic membrane as a reductionist approach to studying the system. Greater clarity over the residues captured in the constructs and which component domains are in use, and why, would also be helpful. The reader is instructed to see methods for detail, but the detail is limited to a tabulated list of constructs. The authors do not specify important details in places, such as why general pTyr antibodies are used for the detection of phosphorylation and which sites are actually being detected by this approach.

6. I encourage the authors to eliminate as much repetition as possible. Some repetition may be important for underscoring important points, however, in many instances, it can leave the reader wondering why the point was restated and whether there is a new idea that they have missed.

7. It would be helpful for the authors to consider precision during revision. For example, the authors state "much more" rather than detailing quantitatively the increase in the latter two paragraphs of page 12.

8. Page 16- what is the impact of the UAA introduction on Btk catalytic activity? What sites were tried and excluded because of yield? There is value in presenting those that didn't work as supplemental data and to illustrate why T403 was considered a success by showing the relative yield on a gel. Why is a 3-component model fit chosen? What are the 3 components?

9. Figure 5 – why are the two pTyr blots not shown as a single blot, rather than as 2 panels from the same blot?

10. End of page 20 – why has the pTyr SH2 domain mutant Grb2 proteins not been tried in the LAT system? It seems to me that the possibility raised in the conclusion could be easily tested with the tools at the author's disposal.

11. Figure 6. 3 delay times are presented and it is not clear why they were chosen. The legend says multiple delay times were used, although it would be more precise to state what exactly they were.

12. Ideas and speculation could be written in a more economical way such that the key messages are more clearly focused.

13. The authors should revise the methods with consideration to whether sufficient detail is presented for another researcher to repeat the work. Currently, the absence of information on vendor, catalog numbers (e.g. for lipids, AzF, antibodies, ULP1 protease), shaking speed, buffer compositions for chromatographic steps, concentrations of antibiotics, speed of centrifugation steps, what room temperature is, what an "appropriate concentration" means, how immunoblot transfers were performed and what type of membrane, and so on, prevents a reader from being able to replicate the work if desired.

Authors appear to be compliant as there is no new code or dataset that needs to be deposited in a repository.

---

## [Author Response]

Essential revisions:1) All reviewers highlighted that antibodies towards specific pTyr sites would be most appropriate as a read-out of phosphorylation because the pan-pTyr antibody could detect ambiguous/off-target signals. Repeating with specific pTyr antibodies is essential.

We attempted repeats with site-specific antibodies and found these antibodies to be less sensitive and lacked the specificity we would hope for in a site-specific antibody. Instead, used mass spectrometry (LC-MS/MS) to confirm that the tyrosine residues in the PLC*γ*2 peptide in the fusion protein were indeed phosphorylated by activated Btk in our experiments. The new mass spectrometry data confirm that we are indeed measuring activity of Btk against physiologically relevant substrate.

2) The variability between duplicate experiments limits robust interpretation throughout the manuscript; it is essential that further independent repeats are performed to ensure the findings are robust.

We have carried out new experiments in which phosphorylation of the PLC*γ*2-peptide fusion and phosphorylation of Btk was measured across a range of Grb2 concentrations to provide a more complete picture of how Btk kinase activity is affected by Grb2. We have also modified our method for quantifying blots to allow for more careful selection of baseline/background intensity, and this method is detailed in the revised Materials and methods section. In addition, we have chosen to explicitly show which intensities correspond to which replicate. This ensures that any comparison made across conditions can be carefully made within each replicate. Although the absolute intensities vary across replicates, the relative intensities are similar. We have also updated the text to reflect that we are making conclusions based on the observed trends and not the intensity values themselves.

3) Figure 6. Further negative control experiments are required: with labeled Btk lacking its PH-TH-PRR domains, immobilised on the membrane with a His tag, as they have done in other experiments; and with the pTyr-binding deficient SH2 domain mutant of Grb2.

We have opted to look at Btk diffusion in the presence or absence of condensed LAT to assess the effect of condensation on Btk diffusion, and we believe the necessary controls have been included to ensure that the change in diffusion is due to condensation and not some other source. Although Btk lacking the PH-TH-PRR domains immobilized via a His tag is a possible additional control, we believe this construct may present additional complications that are beyond the scope of this work. For example, control of the density of His tagged proteins relative to one another on 4% DGS-NTA (Ni) containing supported lipid bilayers has proven to be tricky and sometimes unpredictable. By providing Btk as an additional His tagged protein with LAT and Hck, it may be challenging to ensure that the density of each of these species (and Btk itself) has been maintained. Because the properties of LAT phase transition are highly dependent on LAT density and phosphorylation, we believed this change would make it difficult to compare non-His tagged Btk to His-tagged Btk. However, this could prove a useful tool for future studies after extensive characterization. Prior work has probed the diffusion of LAT itself during condensate formation and it was shown to display very similar overall behavior to what is observed here for Btk across our different conditions (Sun et al., 2022). Prior work has shown that Grb2 R86K is unable to induce LAT condensation and therefore we do not think this would be a particularly useful experiment (Lin et al., 2022).

4) There are suggestions from all 3 reviewers for how the data presentation and writing of the manuscript might be improved. A number of important points relating to data interpretation should also be considered in revision; the novel findings are of interest but data are currently insufficient to support the authors' claims. Extensive editing of the manuscript is required; in its current form, it would not be acceptable for publication.

We have carefully read the reviewers suggestions and the manuscript has been edited extensively for clarity. To limit discussion to the most robust results, we have decided to remove the Western blot data concerning the effects of Grb2 on the Src module of Btk (lacking the PH-TH-PRR modules). We found that the change in signal as a result of Grb2 addition is minimal for this construct, and considerably more experimentation would be required to draw reliable conclusions. Note that these data are not central to the arguments of the paper, and we feel that the manuscript with these data removed is internally consistent.

5) The authors should resolve the discrepant pTyr data in Figure 3 vs 5 Supp2, and present a more complete picture of the analysis of diffusion constants in Figure 5, including estimations of error.

We agree that the data in Figure 5 supplement 2 (numbering from original manuscript) is confusing and have chosen to remove these data from the manuscript. This construct (Btk K49S/R52S, a double-mutant that is defective in PH-TH dimerization) has proven challenging to work within the context of full-length Btk – our previous analysis of this double mutant concerned just the PH-TH module. Estimations of error for diffusion constants (now in Figure 4) and a more complete picture of the analysis (see Materials and methods) are now provided.

Reviewer #2 (Recommendations for the authors):Figure 3. The authors need to perform sufficient replicates to confidently argue that Grb2 directly stimulates Btk activity. Whilst the two replicates of Btk autophosphorylation are relatively convincing, the data supporting stimulated substrate phosphorylation are not.Since substrate phosphorylation should be directly influenced by activation loop autophosphorylation of Btk, these two readouts should be mutually consistent.Furthermore, an inspection of the change in Btk autophosphorylation in Figure 3 panel C is not consistent with that reported for the same time point (5 min) in panel A, suggesting that the authors suffer from a lack of reproducibility in these experiments.

We have carried out additional experiments with the PLC*γ*2-peptide fusion (Figure 3C and Figure 3 – Supplement 2) to show that peptide phosphorylation is also dependent on Grb2 concentration. Additionally, we have updated our representation of these data by showing each replicate separately. We believe this is a better way to represent the data and makes it easier for the reader to interpret. Alternative detection techniques may be more appropriate, but we opted to focus on PLC*γ*2-peptide fusion phosphorylation as we believe this is the most direct measure of Btk catalytic activity.

Figure 4. Again, insufficient replicates are reported to support the authors' claims. In particular, the assertion that Grb2 increases activity of the SH3-SH2-kinase module is based on a single time point (5 min) with two measurements, as the error bars of the other time points overlap between the conditions with and without Grb2. Furthermore, the authors do not observe any significant difference in substrate phosphorylation, which would be an expected consequence of activation loop autophosphorylation.

As noted above, we agree that these data are inconclusive and have opted to remove it from the manuscript.

Figure 5. Missing error bars for diffusion coefficient in panel A (presumably these were the averages of multiple particle tracks?). Again, insufficient replicates for B-C. The conceptual problem here: the authors have designed what they say is a rescue experiment to test whether Grb2 can 'rescue' autophosphorylation of a mutant of Btk impaired in its dimerization on the membrane. However, an inspection of their data reveals no visible defect in Btk autophosphorylation between the wt and mutant proteins (Figure 5 Supplement 2). Furthermore, in order to compare wt with mutant Btk, one would need to know whether the level of Btk phosphorylation detected is sub-stoichiometric or not. Otherwise, Btk may already be maximally phosphorylated with just PIP3-liposomes and an effect of Grb2 would not be detectable. The authors should also comment on why Grb2 is phosphorylated in these experiments, but not in those experiments reported in Figure 3. In the text, the authors reference Figure 3 Supplement 2 as evidence for Grb2 phosphorylation, when in fact there is no sign of Grb2 phosphorylation, and a signal for Grb2 phosphorylation is only found in Figure 5 Supplement 2.

Error bars representing standard deviation have been added for all diffusion coefficients, and explanation of how the fits were obtained has been expanded in the Materials and methods.

The data pertaining to the dimerization-deficient Btk has been removed, we agree that further characterization would be necessary to carefully interpret these data.

As stated above, the apparent phosphorylation of Grb2 is an artifact of the antibody we were using, Phospho-Tyrosine Mouse mAb (pTyr-100) (Cell Signaling, catalog no. 9411). This is shown in Figure 3 – Supplement 2.

Figure 6. To address the concerns with the interpretation of the data presented in this figure, it is recommended to the authors that they perform a negative control experiment with labeled Btk lacking its PH-TH-PRR domains, immobilised on the membrane with a His tag, as they have done in other experiments. The diffusion properties of Btk SH3-SH2-kinase bound to DGS-NTA(Ni) vs full-length Btk bound to PIP3 should be comparable. This control experiment will address whether the reduction in Btk mobility the authors observe is specific to its Grb2-dependent recruitment into LAT condensates, or, whether the condensates create physical barriers to Btk diffusion on the membrane.

As explained in point (3) of the essential revisions above, we do not believe these controls are necessary for the conclusions we are making here, and there are technical reasons that make some of these controls. However, we do believe that further study along these lines would be exciting future work.

The text is unnecessarily long and repetitive, as well as confusing in places, and contains a number of factual errors. The cartoon schematics in multiple figures are often not representative of the experiment being performed.

We have edited the text in its entirety and made many improvements to the writing and figures to make the manuscript more concise, rigorous, and clear.

Reviewer #3 (Recommendations for the authors):Specifically, I recommend:1. In the abstract, the authors have included a citation for another study. This is highly unconventional and I recommend removing it.

We have submitted this manuscript as an *eLife* Reasearch Advance, which requires that we cite the prior study that the current manuscript builds on (Wang et al., 2015).

2. In schematics, the component domains are not always labelled. There's a great deal of assumed knowledge in doing so, but this does not allow the work to be interpreted by a generalist audience, such as eLife's readership.

We have added domain names where appropriate.

3. In the introduction, the Saraste dimer is introduced but the orientation/arrangement of domains is not described or depicted in a way that a non-expert would be able to interpret. The dimer is not shown at all in Figure 1 but should be.

We have added a structure panel to this Figure, Figure 1C.

4. Line 110. Crosslinking suggests a covalent linkage. I urge the authors to reconsider their choice of language.

This language has been updated.

5. Greater clarity on labelling methods in the Results and Discussion would aid interpretation both where TIRF is first mentioned and later where Amber UAA methods are used. In both instances, it would be helpful to the reader to state clearly that these studies use recombinant proteins and a synthetic membrane as a reductionist approach to studying the system. Greater clarity over the residues captured in the constructs and which component domains are in use, and why, would also be helpful. The reader is instructed to see methods for detail, but the detail is limited to a tabulated list of constructs. The authors do not specify important details in places, such as why general pTyr antibodies are used for the detection of phosphorylation and which sites are actually being detected by this approach.

Materials and methods have been updated in greater detail, and further detail for these sections has been included in the Results and Discussion. We have carried out LC-MS/MS to check phosphosites for the PLC*γ*2-peptide fusion. We attempted Western blots with the site-specific antibodies for sites we were interested in, however off-target binding made these antibodies unusable.

6. I encourage the authors to eliminate as much repetition as possible. Some repetition may be important for underscoring important points, however, in many instances, it can leave the reader wondering why the point was restated and whether there is a new idea that they have missed.

We have updated much of the manuscript to make the writing more concise and less repetitive.

7. It would be helpful for the authors to consider precision during revision. For example, the authors state "much more" rather than detailing quantitatively the increase in the latter two paragraphs of page 12.

Where appropriate, we have modified the language to convey magnitude of change.

8. Page 16- what is the impact of the UAA introduction on Btk catalytic activity? What sites were tried and excluded because of yield? There is value in presenting those that didn't work as supplemental data and to illustrate why T403 was considered a success by showing the relative yield on a gel. Why is a 3-component model fit chosen? What are the 3 components?

We did not assess the catalytic activity of the UAA introduction on Btk catalytic activity. T20AzF and H620AzF were also attempted for purification and labeling. Although these variants could be purified, we were concerned that mutation at T20 may have an impact on membrane association (this construct was originally designed for a different purpose), and the H620AzF had more of a tendency to aggregate on supported lipid bilayers.

Initially 3 components were chosen based on the idea that Btk should have at least two diffusive species (monomer and dimer) and fluorescently labeled Btk tends to aggregate on membranes, resulting in a third population that is essentially immobile. After fitting, our data suggest that the monomer and dimer populations cannot be clearly distinguished by this method, and instead there seem to be two very slow-moving populations, one of which is probably the immobile particles. Three components are the lowest number of distributions necessary to fit the data, however, more sophisticated statistical techniques would be needed to assess whether we are over or underfitting with this number of components.

9. Figure 5 – why are the two pTyr blots not shown as a single blot, rather than as 2 panels from the same blot?

This blot was removed for other reasons (cited above), but originally panels are shown in the main figure to save space in the main text, the full blot is shown in the supplement.

10. End of page 20 – why has the pTyr SH2 domain mutant Grb2 proteins not been tried in the LAT system? It seems to me that the possibility raised in the conclusion could be easily tested with the tools at the author's disposal.

This mutant has been shown to be unable to induce LAT phase transition (Lin et al., 2022). Therefore, we decided it would not represent an informative comparison to wild-type Grb2 in this experiment.

11. Figure 6. 3 delay times are presented and it is not clear why they were chosen. The legend says multiple delay times were used, although it would be more precise to state what exactly they were.

An explanation for why these delay times were chosen was expanded in the figure legend and Materials and methods. A fourth delay time was added to show a more complete dataset. Essentially, this method is to ensure that we have not introduced artifacts into the step size distribution by choice of exposure time and step size for single molecules relative to the pixel size of the camera.

12. Ideas and speculation could be written in a more economical way such that the key messages are more clearly focused.

We have reviewed and improved this section of the paper.

13. The authors should revise the methods with consideration to whether sufficient detail is presented for another researcher to repeat the work. Currently, the absence of information on vendor, catalog numbers (e.g. for lipids, AzF, antibodies, ULP1 protease), shaking speed, buffer compositions for chromatographic steps, concentrations of antibiotics, speed of centrifugation steps, what room temperature is, what an "appropriate concentration" means, how immunoblot transfers were performed and what type of membrane, and so on, prevents a reader from being able to replicate the work if desired.

We have added much more detail to Materials and methods to address this concern.